# Comprehensive prediction of secondary metabolite structure and biological activity from microbial genome sequences

Michael A. Skinnider [1,2,3,4 ✉], Chad W. Johnston[1,2,3,5], Mathusan Gunabalasingam[1,2], Nishanth J. Merwin[1,2], Agata M. Kieliszek[3], Robyn J. MacLellan[3], Haoxin Li[3], Michael R. M. Ranieri[1,2], Andrew L. H. Webster[1,2], My P. T. Cao[1,2], Annabelle Pfeifle [3], Norman Spencer [3], Q. Huy To[1,2], Dan Peter Wallace[3], Chris A. Dejong [3 ✉] & Nathan A. Magarvey[1,2]

Novel antibiotics are urgently needed to address the looming global crisis of antibiotic resistance. Historically, the primary source of clinically used antibiotics has been microbial secondary metabolism. Microbial genome sequencing has revealed a plethora of uncharacterized natural antibiotics that remain to be discovered. However, the isolation of these molecules is hindered by the challenge of linking sequence information to the chemical structures of the encoded molecules. Here, we present PRISM 4, a comprehensive platform for prediction of the chemical structures of genomically encoded antibiotics, including all classes of bacterial antibiotics currently in clinical use. The accuracy of chemical structure prediction enables the development of machine-learning methods to predict the likely biological activity of encoded molecules. We apply PRISM 4 to chart secondary metabolite biosynthesis in a collection of over 10,000 bacterial genomes from both cultured isolates and metagenomic datasets, revealing thousands of encoded antibiotics. PRISM 4 is freely available as an interactive web application at http://prism.adapsyn.com.

---

[1] Department of Biochemistry & Biomedical Sciences, Michael G. DeGroote Institute for Infectious Disease Research, McMaster University, Hamilton, ON, Canada. [2] Department of Chemistry & Chemical Biology, Michael G. DeGroote Institute for Infectious Disease Research, McMaster University, Hamilton, ON, Canada. [3] Adapsyn Bioscience, Hamilton, ON, Canada. [4] Michael Smith Laboratories, University of British Columbia, Vancouver, BC, Canada. [5] Institute for Medical Engineering and Science, Massachusetts Institute of Technology, Cambridge, MA, USA. ✉email: michaelskinnider@gmail.com; chris_dejong@adapsyn.com

The overwhelming majority of antibiotics currently in clinical use are derived from naturally occurring small molecules produced by microbes[1]. The biosynthetic pathways responsible for the production of these molecules have been honed over long evolutionary time scales in order to provide microbes with competitive advantages in their natural environments[2]. These pathways are encoded within the genomes of the producing organisms, and comparative genomics studies have suggested a wealth of novel antibiotics encoded in the genomes of both culturable and unculturable organisms that remain to be discovered[3–5]. Directed discovery of these unknown antibiotics, guided by genome sequencing data, could provide a means to address the growing clinical need for new antibiotics to combat drug-resistant pathogens[6].

With the amount of microbial genome sequence information deposited in public databases continuing to increase at an exponential rate (Supplementary Fig. 1), methods to leverage this data towards antibiotic discovery are urgently needed. However, whereas a plethora of methods are available to identify the genomic loci responsible for natural antibiotic biosynthesis[7–9], few tools exist to link these loci to the specific chemical structures of their encoded products. The challenges inherent to the latter task far exceed those involved in genome annotation: nature employs a dizzying array of enzymatic catalysts to construct structurally complex molecules from simple building blocks. Moreover, these catalysts are arranged in multi-gene clusters that can be categorized into dozens of distinct families. Existing tools can generate predictions of genomically encoded natural antibiotic structures from small regions of this vast biosynthetic space[7,10], but a comprehensive platform is lacking.

We previously described PRISM, a genome analysis toolkit and web application to predict the complete chemical structures of genomically encoded nonribosomal peptides and polyketides[11]. However, despite subsequent extension to families such as ribosomally encoded and posttranslationally modified peptides (RiPPs)[12], PRISM's coverage of industrially important chemical space remained incomplete. Here, we present PRISM 4, which enables genome-guided chemical structure prediction for every class of bacterial natural antibiotics currently in clinical use, including aminoglycosides, nucleosides, β-lactams, alkaloids, and lincosamides among other classes of metabolites. Moreover, PRISM 4 achieves a dramatic increase in coverage of enzymatic tailoring reactions encoded within canonical thiotemplated pathways (Fig. 1 and "Methods"). PRISM achieves accurate structure prediction by connecting biosynthetic genes to the enzymatic reactions they catalyze, permitting the in silico reconstruction of complete biosynthetic pathways (Supplementary Figs. 2 and 3) as well as their final products (Fig. 1a, b). In total, PRISM 4 includes 1772 hidden Markov models (HMMs) and implements 618 in silico tailoring reactions in order to predict the chemical structures of 16 different classes of secondary metabolites, making it a comprehensive resource to link microbial genome sequence information to the natural antibiotics encoded within (Fig. 1c, Supplementary Table 1, and Supplementary Data 1).

## Results

### PRISM 4 generates accurate structure predictions for known BGCs.

To evaluate the accuracy of PRISM 4, we assembled a comprehensive set of 1281 biosynthetic gene clusters (BGCs) with known products from public databases and extensive literature curation, subject to multiple rounds of manual review by a team of natural products chemists to correct errors in chemical structures or the boundaries of deposited nucleotide sequences (Methods). PRISM 4 detected 1230 of these reference BGCs

(96%), representing an increase of 40% over the original PRISM release, as well as a slight increase in sensitivity over antiSMASH 5, which detected 1212 (Fig. 2a). Moreover, PRISM 4 generated at least one predicted chemical structure for 1157 of the 1230 detected BGCs (94%), an increase of at least 54% over antiSMASH 5 or NP.searcher, which predicted structures for 753 and 398 BGCs, respectively (Fig. 2b). To quantify the similarity of predicted structures to the true cluster products, we calculated the Tanimoto coefficient[13] (Tc) between real and predicted structures from each cluster, a measure of chemical similarity that reflects the fraction of substructures shared between the two molecules, and compared these to predicted and true structures from random BGCs pairs (Methods). Using this metric, we found PRISM 4 achieved statistically significant predictive accuracy across a wide range of secondary metabolite classes (Fig. 2e). For the subset of 385 BGCs with structure predictions generated by all four programs, we compared the Tc between true products and predicted structures from PRISM 4, antiSMASH 5, and NP. searcher, finding that PRISM 4 was significantly more accurate in both comparisons (both $p < 10^{-15}$, paired Brunner–Munzel test; Fig. 2c and Supplementary Data 2); pairwise comparisons were likewise highly significant ($n = 398$ and 753, respectively, both $p < 10^{-15}$; Supplementary Fig. 4). We additionally quantified the accuracy of structure predictions based on the functional groups they contained[14]. Using the Jensen-Shannon divergence to compare the distributions of functional groups found in true and predicted structures, we observed that the functional group content of PRISM 4 predicted structures was significantly more similar to that of true products than that of structures predicted by antiSMASH 5 or NP.searcher (bootstrap $p < 0.001$; Fig. 2d).

In some cases, the precise substrate of the reaction catalyzed by a given enzyme is not unambiguously predictable from protein sequence alone: for instance, a halogenase may catalyze chlorination at a number of different sites within a molecule. For this reason, PRISM considers all possible sites of each tailoring reaction, and combinations thereof, when generating predicted structures. To validate this strategy, we compared the median and maximum Tc between predicted and true structures for each cluster, finding the maximum Tc to be significantly greater ($p < 10^{-15}$; Fig. 2e and Supplementary Fig. 5). We also quantified the size of the combinatorial search space for each family of metabolites (Supplementary Fig. 6), finding that the majority of classes could usually be predicted within a dozen or fewer combinatorial plans, but a subset of families were associated with a greater degree of structural uncertainty (most notably aminoglycosides, in which the configurations of the glycosidic bonds cannot be predicted from primary sequence).

### PRISM 4 predicts natural product-like products for cryptic BGCs.

To gain a broader perspective on PRISM 4's ability to predict encoded metabolite structures from genome sequence, we used PRISM 4 to analyze secondary metabolism in a collection of 3,759 dereplicated complete bacterial genomes[15]. For this comparison, we focused on PRISM and antiSMASH, as platforms designed to analyze BGCs from a wide range of biosynthetic families. Among 22,446 identified clusters, PRISM 4 generated at least one predicted structure for 7404, a significantly greater proportion than antiSMASH 5 ($p < 10^{-15}$, $\chi^2$ test), with 3184 clusters having structures predicted only by PRISM 4, compared to 500 only by antiSMASH 5 (Fig. 3 and Supplementary Data 3a). Notably, PRISM 4 predicted hundreds of complete chemical structures for families of metabolites such as β-lactams, alkaloids, phosphonates, cyclodipeptides, bisindoles, and aminoglycosides, for which antiSMASH 5 predicted only a handful of structures, or none at all (Fig. 3a). PRISM 4 also generated the majority of

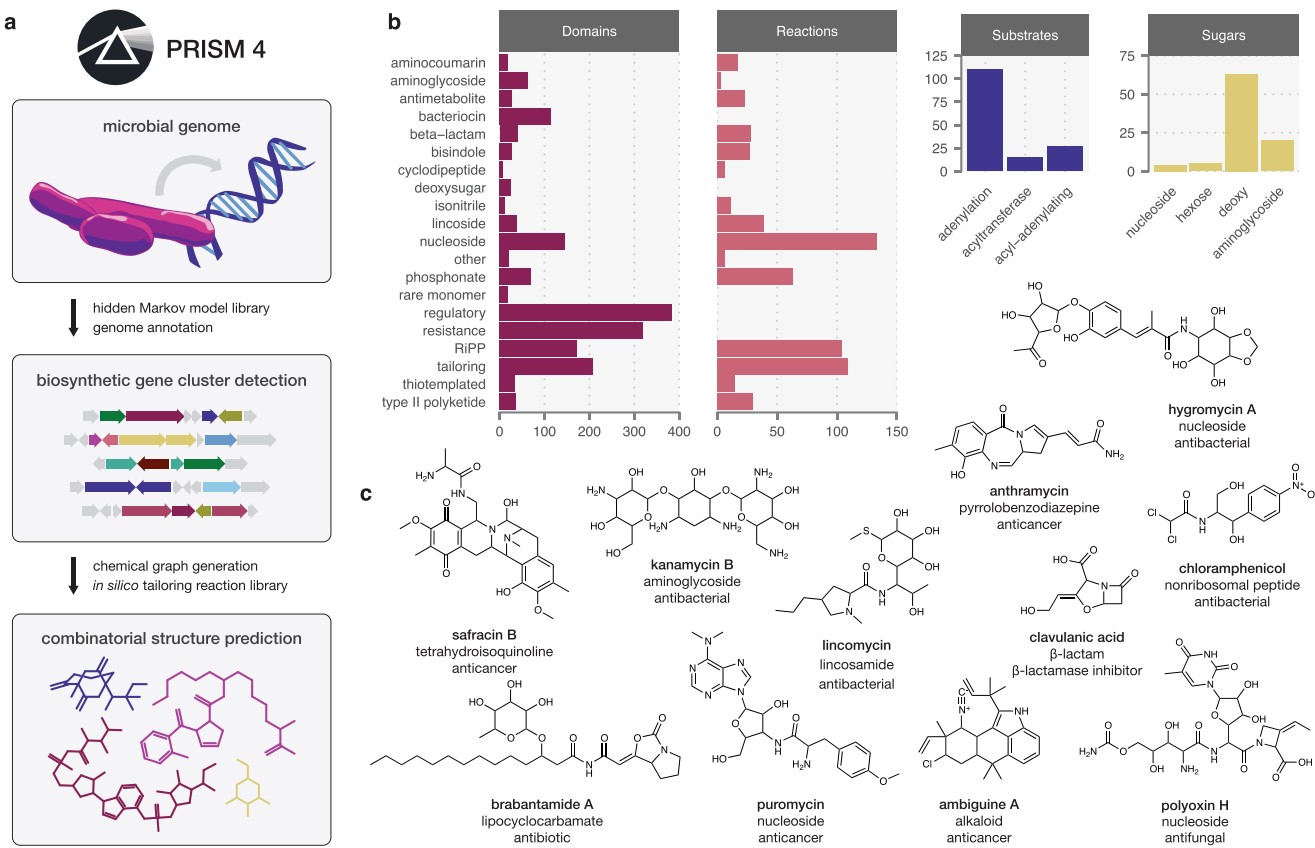

**Fig. 1 A comprehensive platform for genome-guided prediction of secondary metabolite chemical structures. a** Schematic overview of PRISM 4. Microbial genome sequences are annotated using a library of 1,772 HMMs, and secondary metabolite BGCs are identified using a rule-based approach. Combinatorial, graph-based chemical structure prediction is effected using a library of 618 virtual tailoring reactions. **b** Total number of HMMs, virtual tailoring reactions, substrates, and sugars incorporated in PRISM 4. **c** Examples of predicted chemical structures generated by PRISM 4 for newly added families of secondary metabolites. Source data are provided as a Source Data file.

structure predictions for several bacterial phyla whose biosynthetic capacity has historically not been widely appreciated, such as Desulfobacterota, Spirochaetota, or Campylobacterota (Fig. 3b). Given that phylogenetically distinct organisms are more likely to produce novel products[4,16], this finding suggests PRISM 4 may have particular utility for genome mining of molecules with scaffolds or activities that diverge from those present in well-studied organisms.

Because the true structures of the metabolites encoded by these loci are not known, we were unable to directly assess the accuracy of structure prediction. Instead, we asked whether predicted structures had structural features characteristic of known natural products[17]. Previous studies have found that a relatively small proportion of natural products are within the chemical space defined by Lipinski's "rule of five," a set of guidelines developed to facilitate the design of orally bioavailable drugs[18,19]. Relative to structures predicted by antiSMASH 5, a lower proportion of PRISM 4 predictions were within Lipinski's rule of five space[20] ($\chi^2$ test, $p < 10^{-15}$; Fig. 3c), having greater molecular weights (paired Brunner–Munzel test, $p < 10^{-15}$; Fig. 3d), more hydrogen bond donors and acceptors ($p < 10^{-15}$; Supplementary Fig. 7a, b), and greater octanol-water partition coefficients ($p < 10^{-15}$; Supplementary Fig. 7c). PRISM 4 predictions were also more structurally complex, as quantified using the Bertz topological complexity index[21] ($p < 10^{-15}$; Fig. 3e), a measure of molecular complexity that incorporates both the complexity of the bonding and the distribution of heteroatoms. Moreover, PRISM 4 predictions were also more structurally diverse, as quantified by the median intra-set Tc ($p < 10^{-15}$; Fig. 3f). Finally, PRISM 4

predictions displayed a greater degree of structural similarity to known natural products, as quantified either by their median Tc to the set of known natural products in Natural Products Atlas ($p < 10^{-15}$; Fig. 3g), or by their 'natural product-likeness' score[22] ($p < 10^{-15}$; Supplementary Fig. 7d). Taken together, these results indicate PRISM generates complex, diverse, and natural product-like chemical structure predictions from large genomic datasets.

To evaluate the BGC detection functionality of PRISM and antiSMASH, we carried out a blinded review of 200 randomly sampled clusters detected only by one of the two methods. Manual annotation suggested up to 55% of antiSMASH-only BGCs represented false positives (FPs), compared to up to 37% of PRISM-only BGCs ($p = 0.016$, $\chi^2$ test; Supplementary Fig. 8). Among antiSMASH-only BGCs, recurrent categories of FPs included minimal fatty acid synthases, DUF692-associated bacteriocins, putative phosphonate BGCs associated with cell wall biosynthesis machinery, and isolated prenyltransferases classified as terpene BGCs. It should be noted that a trade-off between specificity and sensitivity is inherent to any prediction task, and the higher rate of FPs for antiSMASH also expectantly affords it a greater ability to detect—though not to predict structures for—novel or divergent BGC types.

**PRISM 4 enables chemical structure prediction from metagenomic data.** Rapid progress in metagenomic sequencing technologies, accompanied by rapid advances in computational approaches for genome assembly from metagenomic data[23], has revealed a wealth of undiscovered antibiotics within uncultured organisms[3]. We used PRISM 4 to analyze secondary metabolism

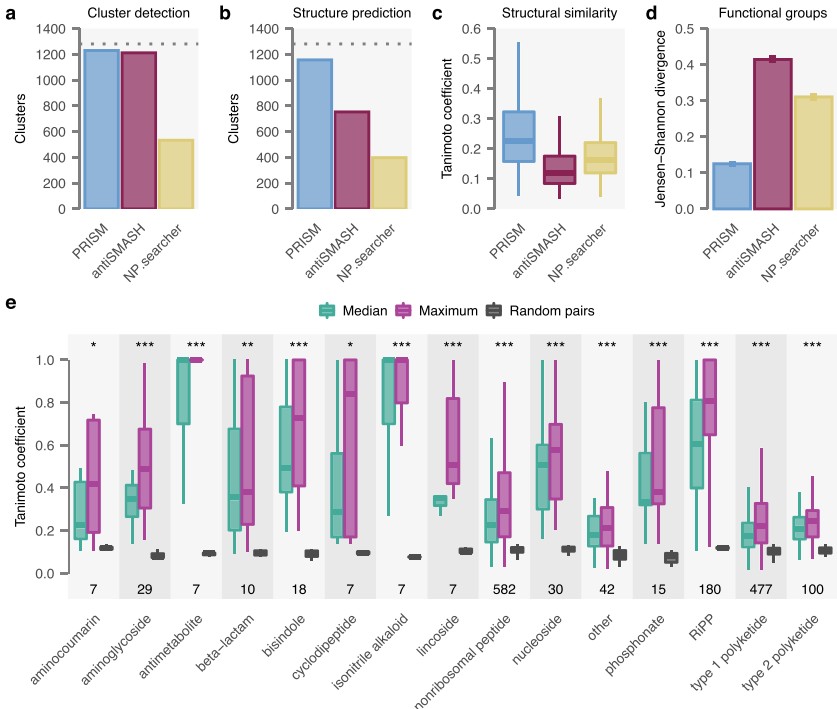

**Fig. 2 PRISM 4 generates highly accurate chemical structure predictions. a** Number of BGCs within a manually curated gold standard set ($n = 1{,}281$; dotted line) identified by PRISM 4, antiSMASH 5, and NP.searcher. **b** Number of BGCs within the gold standard set with at least one structure predicted by each program. **c** Median Tanimoto coefficient between true and predicted structures for the subset of gold standard BGCs with at least one predicted structure generated by all four programs ($n = 385$). **d** Jensen–Shannon divergence between functional group content of true and predicted structures for each program. Errors bars show standard deviation of bootstrap resampling. **e** Median and maximum Tanimoto coefficients between true and predicted structures generated by PRISM 4 for the gold standard set, by biosynthetic family, and compared to the median Tanimoto coefficient between predicted structures and non-matched BGCs ("random pairs"). Top, statistical significance of the comparison between median and random Tanimoto coefficients (***$p < 0.001$; **$p < 0.01$; *$p < 0.05$, two-sided $t$-test). Bottom, number of BGCs from each family in the gold standard set (n). Box plots show median (horizontal line), interquartile range (hinges), and the smallest and largest values no more than 1.5 times the interquartile range (whiskers) throughout. Source data are provided as a Source Data file.

in a collection of 6,362 dereplicated metagenome-assembled genomes (MAGs)[15,23]. PRISM 4 generated predicted structures for 2630 of 10,814 clusters, representing the vast majority of structure predictions for this collection of genomes (~96%), and significantly more than antiSMASH 5 ($p < 10^{-15}$, $\chi^2$ test; Supplementary Fig. 9a, b and Supplementary Data 3b). In addition to well-studied classes of metabolites, notably those originating from thiotemplated assembly lines (nonribosomal peptides and polyketides) as well as ribosomally synthesized and posttranslationally modified peptides (RiPPs), we found biosynthesis of phosphonate-containing natural products to be surprisingly common among uncultured organisms (Supplementary Fig. 9a). PRISM 4 metagenomic structure predictions also possessed structural features characteristic of known natural products, including a lower proportion in rule-of-five space, larger molecular weights, greater topological complexity, increased internal diversity, and greater similarity to known natural product structures than a matched set of structure predictions from antiSMASH 5 (Supplementary Fig. 9c–g). Collectively, these results reinforce the notion that a wealth of biologically active metabolites are encoded within the genomes of uncultured organisms, and highlight the value of PRISM 4 for interrogation of antibiotic biosynthesis in large metagenomic datasets.

**Quantitative structure-activity relationships of cryptic molecules.** Taken together, these analyses indicate PRISM 4 generates realistic structure predictions from the genomes of diverse cultured and uncultured organisms, with a high degree of chemical similarity to true products in the case of known BGCs, and

structural features characteristic of known secondary metabolites in the case of cryptic BGCs discovered by genome mining. We therefore asked whether these high-quality predicted structures could be leveraged to address another key challenge in genome-guided discovery of natural antibiotics: namely, prioritizing particular BGCs or producing organisms with the greatest likelihood of producing biologically active metabolites for targeted discovery. We undertook an extensive literature review to systematically curate bioactivity data for the 1281 BGCs in the gold standard set, and trained support vector machines (SVMs) to predict the probability that a given BGC produces a compound with antibacterial, antifungal, antiviral, antitumor, or immunomodulatory activity, using tenfold cross-validation to evaluate model accuracy. To evaluate the performance of these models, we calculated the area under the receiver operating characteristic curve (AUC), and compared the observed AUCs to those expected from random predictors[24]. In all cases, these models yielded significantly more accurate predictions of biological activity than random expectation (all $p < 10^{-15}$, Wilcoxon rank-sum test; Fig. 4a). Furthermore, classifiers trained on the chemical fingerprints of PRISM predicted structures were significantly more accurate than classifiers trained on Pfam domains, with a mean increase of 7.5% in the AUC ($p < 10^{-15}$, Fisher integration of DeLong tests; Fig. 4a). This increase in performance supports the notion that chemical structure prediction is essential to high-accuracy prediction of the biological activity of genetically encoded metabolites. We refer to this approach as quantitative predicted structure-activity relationship modeling, or QPSAR.

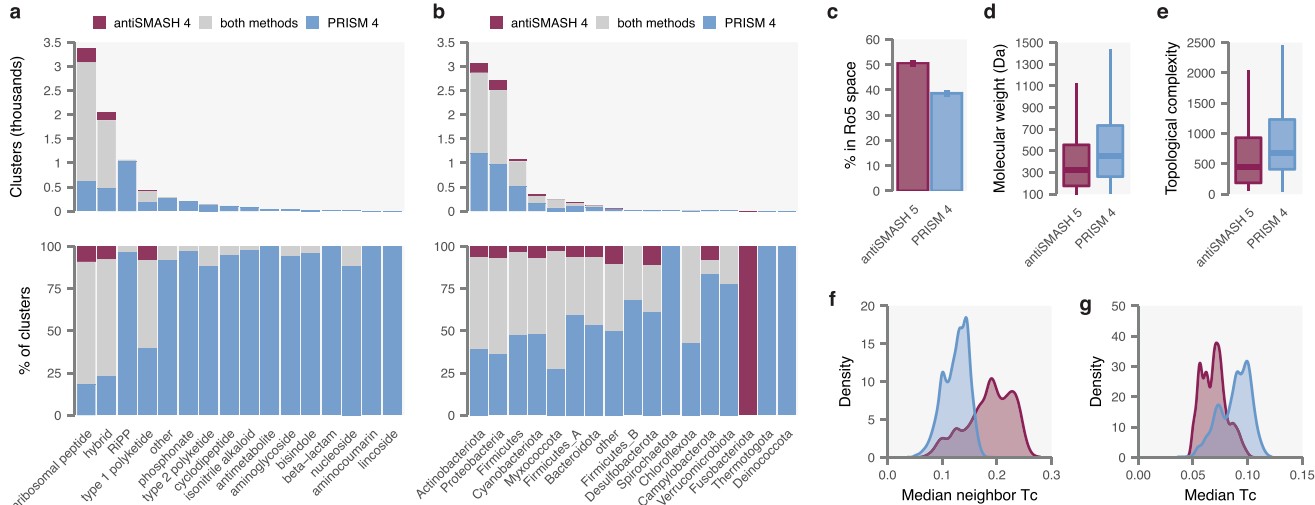

**Fig. 3 PRISM 4 reveals secondary metabolite biosynthesis in 3,759 complete bacterial genomes. a**, **b** Number of BGCs with at least one chemical structure predicted by PRISM 4, antiSMASH 5, or both methods in a collection of 3,759 dereplicated complete bacterial genomes, by biosynthetic family (**a**) and phylum of producing organisms (**b**), as classified in the Genome Taxonomy Database (GTDB)[15]. **c**–**g** Structural features of $n = 4220$ pairs of predicted secondary metabolites from BGCs with products predicted by both PRISM 4 and antiSMASH 5. **c** Percent of predicted structures in Lipinski rule of five space[20]. Error bars show the standard error of the sample proportion. **d** Molecular weight of predicted structures. **e** Bertz topological complexity index[21] of predicted structures. **f** Internal diversity of predicted structures, as quantified by median Tanimoto coefficient to all other predicted structures in the set. **g** Similarity of predicted structures to known natural products, as quantified by the median Tanimoto coefficient to the set of known natural products in the Natural Products Atlas. Box plots show median (horizontal line), interquartile range (hinges), and the smallest and largest values no more than 1.5 times the interquartile range (whiskers) throughout. Source data are provided as a Source Data file.

We next used the trained QPSAR models to systematically discover biosynthetic loci responsible for the production of bioactive metabolites within the complete collection of over 10,000 complete or metagenome-assembled bacterial genomes. At a false discovery rate of 10%, PRISM 4 identified 1589 BGCs producing antibacterial compounds, 331 antiviral BGCs, 289 immunomodulatory BGCs, 272 antifungal BGCs, and 248 antitumor BGCs, in addition to a further 1055 BGCs with more than predicted biological activity (Fig. 4b). To obtain a global overview of the chemical diversity within this dataset, we applied the non-linear dimensionality reduction technique UMAP (uniform manifold approximation and projection)[25] to the chemical fingerprints of PRISM predicted structures. Unlike some other non-linear dimensionality reduction methods, UMAP approximately preserves global structure, meaning points that are close in the low-dimensional space are also close in the high-dimensional space, and vice-versa. This visualization of the complete predicted chemical space revealed substantial chemical diversity within each bioactivity class (Fig. 4c). Notably, predicted structures from complete and MAGs were evenly distributed across the manifold (Fig. 4d), suggesting that potential differences in the quality or completeness of MAGs[26] do not necessarily preclude realistic structure prediction. We also asked whether the MAGs, recovered predominantly from environmental and non-human gastrointestinal samples[23], were enriched or depleted for the production of metabolites with specific biological activities, relative to the set of complete genomes. Intriguingly, this comparison revealed a marked enrichment for biosynthesis of immunomodulatory agents within the latter set ($\chi^2$ test, $p = 3.7 \times 10^{-8}$; Fig. 4e), suggesting a particularly underappreciated diversity of this class of metabolites beyond well-studied microbes. Collectively, these results highlight the importance of chemical structure prediction in deriving accurate models of biological activity for cryptic biosynthetic loci, and provide a roadmap for targeted discovery of thousands of antibiotic, antitumor, and immunomodulatory compounds encoded within sequenced bacterial genomes.

## Discussion

Early microbial genome sequencing projects revealed dozens of cryptic biosynthetic loci within the genomes of well-studied, industrially important microorganisms, spurring predictions that genome mining would usher in a second 'golden age' of antibiotic discovery. Yet, despite notable successes, the impact of genomics on natural antibiotic discovery has been considerably more modest than originally anticipated. Although it is now straightforward to identify clusters of genes responsible for secondary metabolite biosynthesis, translating between genome sequence and the complete chemical structures of the natural antibiotics encoded therein represents a key challenge, and one that has taken on an increasing importance in an era of growing global antibiotic resistance. PRISM 4 represents the most comprehensive effort to address this challenge to date. Our analyses of the natural molecules encoded within thousands of sequenced genomes uncover a vast undiscovered landscape of evolved chemistry. We show that these predicted chemical structures can further be leveraged to develop accurate models of biological activity, which we use to identify thousands of antibiotic, antitumor, and immunomodulatory agents. We make this resource freely available to spur discovery at http://prism.adapsyn.com.

Some limitations should be noted. In developing PRISM 4, we set out to codify an enormous corpus of knowledge, accumulated over decades of research in biosynthesis and enzymology, into an algorithmically tractable form. An inevitable consequence of this approach is that PRISM relies on homology between newly detected proteins and known enzymatic machinery in order to reveal BGCs and predict the structures of their genetically encoded products. For this reason, PRISM can neither identify BGCs from undescribed families, nor predict novel enzymatic activities. More generally, current models of secondary metabolite biosynthesis are incomplete, which places an inherent limit on the accuracy of structure prediction; we have sought to address this by revising the systems used for BGC detection and structure prediction as additional information has become available.

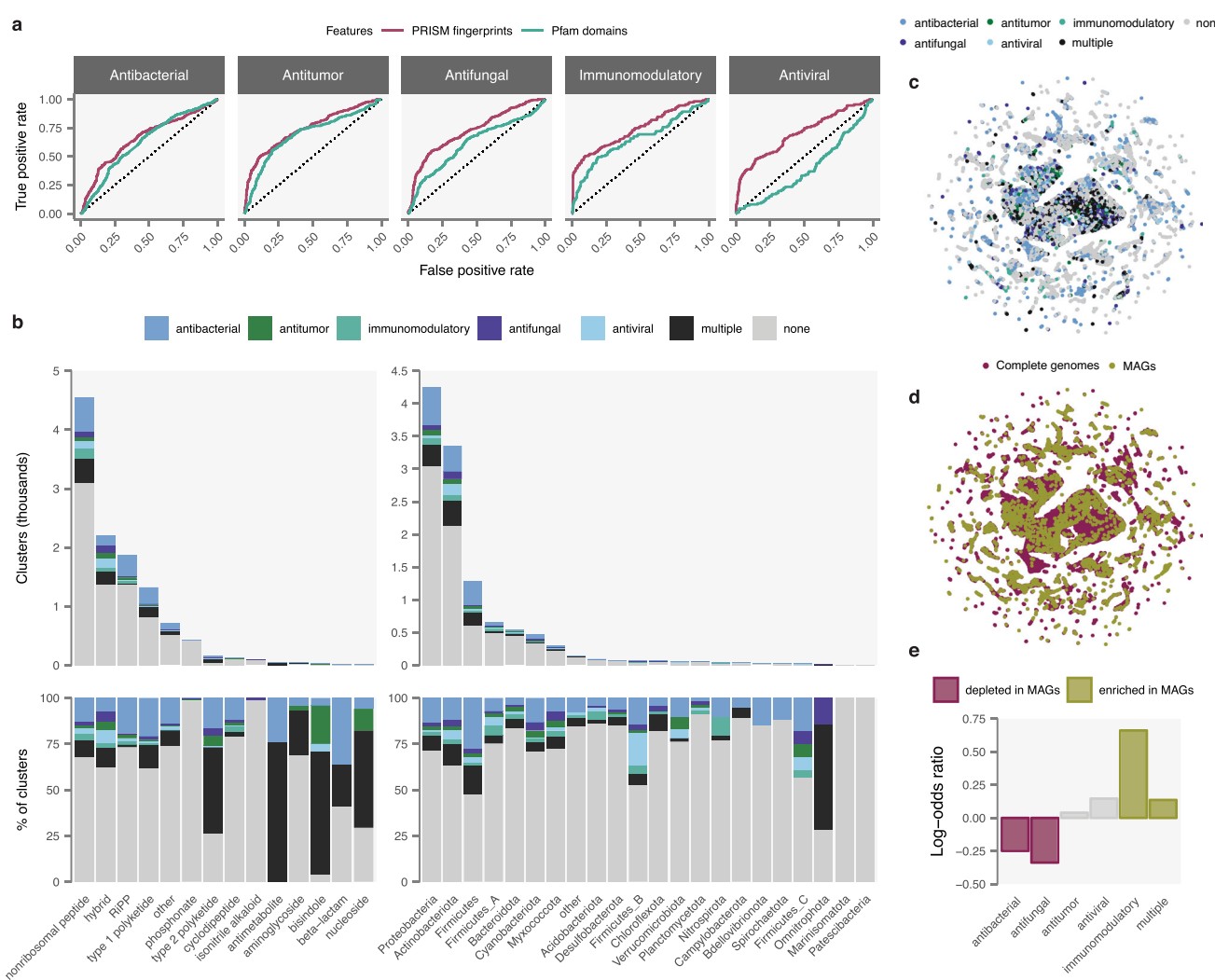

**Fig. 4 Quantitative predicted structure-activity relationship (QPSAR) modeling reveals thousands of genomically encoded antibiotics. a** Receiver operating characteristic (ROC) curves for support vector machine (SVM) models trained on Pfam domains found within biosynthetic gene clusters or chemical fingerprints of PRISM predicted structures. **b** Distribution of BGCs predicted to produce secondary metabolites with antibacterial, antitumor, immunomodulatory, antifungal, antiviral, multiple, or no biological activities in a collection of 10,121 complete or metagenome-assembled prokaryotic genomes, by biosynthetic family (left) or producing organism phylum (right), as classified in the Genome Taxonomy Database (GTDB)[15]. **c, d** Visualization of predicted structure chemical space by uniform manifold approximation and projection (UMAP)[25], colored by biological activity (**c**) or genome origin (**d**). **e** Enrichment or depletion of secondary metabolites by predicted biological activity in metagenome-assembled genomes (MAGs), relative to complete bacterial genomes. Source data are provided as a Source Data file.

Recently, we and others have shown that deep learning-based methods can enable more flexible and accurate detection or characterization of BGCs or individual biosynthetic components[27,28]. However, at present these approaches still rely on interfacing with rule-based systems such as that employed by PRISM 4 to permit structure prediction[27], or else are not capable of generating predicted structures[28]. In the future, more sophisticated machine-learning approaches might enable the end-to-end prediction of encoded small molecules directly from primary sequence. Finally, PRISM 4 was designed primarily for prokaryotic genome analysis and thus cannot identify BGCs families thought to be specific to eukaryotes, and—like all tools for genome annotation—may produce incongruous results when applied to fragmented or low-quality genome assemblies.

## Methods

**Overview of PRISM 4**. PRISM 4 is a cloud-based, interactive web application, with a back-end written in the Java programming language. The web application itself

consists of a VueJS front-end, paired with a Python API that distributes submissions to background workers, and is available at http://prism.adapsyn.com. A number of steps have been taken to ensure the high performance of the web application, including horizontal distribution of individual PRISM runs over the cloud, as well as optimization of key bottlenecks to reduce the runtime by approximately an order of magnitude over PRISM 3 (ref. [29]). Here, we provide a brief overview of the PRISM workflow and the essential changes that distinguish PRISM 4 from previous versions. In Supplementary Note 1, we provide a comprehensive description of the web server, including the user interface and output, a complete description of the methodology underlying BGC detection and chemical structure prediction, and the approaches taken to extend structure prediction to additional BGC families or expand existing ones within PRISM 4. In brief, PRISM 4 takes as input a DNA sequence in FASTA or GenBank format, then queries open reading frames (ORFs) identified therein against a library of 1772 HMMs, complemented by collections of BLAST databases, conserved protein motifs, and machine-learning classifiers, to identify enzymatic domains involved in secondary metabolite biosynthesis and, in some cases, assign them to subtypes, infer their substrates, or otherwise predict their activity. BGCs are identified using a rule-based approach, generally requiring two or more biosynthetic domains to be found in close genomic proximity to reduce the rate of FPs[12].

The biosynthetic information identified from DNA sequence in this manner is subsequently used to predict complete chemical structures for the encoded product

(s) of each BGC. A key challenge in this process is to address cases where the precise substrate of a reaction catalyzed by a given enzyme is not unambiguously predictable from protein sequence alone. As an illustrative example, reactions catalyzed by phosphotransferases or sulfotransferases can generally occur at any free hydroxyl within a molecule. PRISM 4 takes a combinatorial, graph-based approach to structure prediction, with the goal of enumerating all possible products of the identified set of biosynthetic domains. Under this paradigm, the complete biosynthetic pathway and its product are modeled as a series of transformations of a chemical graph, which itself comprises a set of chemical subgraphs. These subgraphs are inferred based on the enzymatic content of the BGC. Each subgraph represents an individual residue, such as a nucleotide or proteinogenic amino acid, or combination of residues with a fixed pattern of connectivity, such as ketide units activated by adjacent modules in a polyketide synthase. The chemical graph is then derivatized based on an in silico knowledgebase of 618 virtual tailoring reactions, each of which links a single enzyme to the reaction it catalyzes. A tailoring reaction involves a series of bond order changes (including bond addition or removal) and atom removal, though never atom addition, to the chemical graph of a biosynthetic intermediate. All 618 reactions are implemented as Java classes, rather than as pattern-based transformations such as the SMIRKS notation, affording a great deal of flexibility in reaction modeling. An example of the distinction between chemical subgraphs and tailoring reactions is depicted in Supplementary Fig. 2, in which the isonitrile geranyltranferase FamD2 activates geranyl pyrophosphate (a chemical subgraph), then catalyzes geranylation of an indole ring (a tailoring reaction). Finally, the complete set of potential biosynthetic pathways, or a large random sample thereof, is inferred when any ambiguity is present in either the chemical graphs or reactions associated with a given BGC. Modeling biosynthesis as a series of tailoring reactions executed on a set of chemical subgraphs allows PRISM 4 to faithfully represent complete biosynthetic pathways, as illustrated in Supplementary Figs. 2 and 3 for two exemplary natural antibiotics. Finally, PRISM generates rich interactive web pages as output, including HTML5-based graphics, to assist the user in exploring the results.

Previous versions of PRISM introduced complete chemical structure prediction for select classes of secondary metabolites, most notably nonribosomal peptides and polyketides and RiPPs[11,12,29,30]. However, coverage of pharmaceutically and industrially relevant secondary metabolite classes remained incomplete. We undertook a comprehensive effort to develop genome-guided chemical structure prediction functionality for all biosynthetic classes of bacterial natural antibiotics that are currently in clinical use. We developed libraries of 145 HMMs and 133 virtual tailoring reactions to predict chemical structures for 21 subtypes of nucleoside natural products; libraries of 40 HMMs and 28 virtual tailoring reactions to predict structures for 11 subtypes of β-lactams, including both β-lactam antibiotics and β-lactamase inhibitors; libraries of 39 HMMs and 39 virtual tailoring reactions to predict lincosamide structures; and libraries of 12 HMMs and 11 virtual tailoring reactions to identify and predict isonitrile alkaloid structures. In addition, we developed a library of 63 HMMs, and revised and extended our previously described algorithm for deoxy sugar prediction[31], in order to predict aminoglycoside chemical structures. The complete sets of HMMs and virtual tailoring reactions are enumerated in Supplementary Data 1, and further detail on these additional classes is provided in Supplementary Note 1. Compared to PRISM 3, PRISM 4 includes a total of 1083 newly developed HMMs (an increase of 145%) and 334 new reactions (an increase of 118%) that were not included in previous versions (Supplementary Fig. 10).

We also extended structure prediction functionality for existing biosynthetic families within PRISM. RiPP structure prediction was augmented by the addition of three additional families, and refinement of some existing HMMs or reactions on the basis of updates to the current understanding of RiPP biosynthesis. More significantly, we undertook a systematic effort to expand structure prediction for canonical thiotemplated (nonribosomal peptide and polyketide) BGC products. First, we identified specific chemotypes that were poorly predicted within PRISM, such as pyrrolobenzodiazepines, tetrahydroisoquinolines, or lipocyclocarbamates. Second, we identified unusual monomers that were not accounted for in PRISM, such as homotyrosine, tambroline, or aziridine-containing amino acids. Finally, we incorporated limited structure prediction functionality for two additional minor classes of secondary metabolites (phenazines and isopropylstilbenes), and expanded the scope of BGC detection to include bacteriocins and nonribosomal peptide synthetase (NRPS)-independent siderophores. Complete details of the updates to PRISM functionality are provided in Supplementary Note 1. In total, PRISM 4 can predict complete chemical structures for 17 biosynthetic families and identify BGCs for a further 11 (Supplementary Table 1).

**Validation of PRISM 4 structure predictions**. In order to validate PRISM 4 structure predictions, we undertook the curation of a comprehensive 'gold standard' database of 1281 prokaryotic BGCs, linked to known secondary metabolites with unambiguously assigned chemical structures. We compiled BGCs from a number of existing databases, including MIBiG[32], ClusterMine360[33], DoBIS-CUIT[34], and NRPS-PKS[35]. These were complemented by extensive manual searching of the NCBI database to retrieve known BGCs observed to be absent from any of these resources, as well as from our own in-house database. We further created a series of PubMed alerts, using a number of terms related to secondary

metabolite biosynthesis, and manually reviewed articles on a weekly basis to identify newly published BGCs.

During the course of investigation, we noticed that many of the chemical structures associated with known BGCs in public databases contained errors. These ranged from minor issues likely introduced by cheminformatics software (e.g., representation of amide bonds by their imidic acid tautomers[36]) to more major structural issues (e.g., structural errors affecting large moieties of the product, or even entirely incorrect products associated with a BGC). Further, a large number of known BGCs did not have an associated chemical structure. We therefore took a systematic review of all chemical structures in order to ensure the accuracy of our 'gold standard' dataset. Each structure associated with a gold standard BGC was subject to two independent rounds of manual review by natural products chemists, with any remaining discrepancies resolved by consensus.

A parallel review of the BGC nucleotide sequences was performed and a number of issues were identified, consisting primarily of cases where contigs larger than the BGC itself were deposited in public databases by the original authors. However, we also identified cases where incomplete BGCs had been deposited, as well as BGCs that spanned more than one contig in originally deposited assemblies. The nucleotide sequences of these BGCs were corrected using publicly available genome assemblies when possible, and omitted from the final dataset otherwise. As a final step, we used a combination of nucleotide BLAST and manual review to assign the taxonomy of the producing organism for each BGC, and removed a small number of remaining BGCs from eukaryotic organisms.

In total, this process led to the curation of a dataset of 1281 BGCs associated with 1,434 molecules. 125 BGCs were associated with more than one molecule. To quantitatively assess the accuracy of PRISM 4 chemical structure predictions with reference to these known products, we calculated the Tanimoto coefficient (Tc) between the chemical fingerprints for each pair of true and predicted structures[13]. The ECFP6 chemical fingerprint[37], with a length of 1024 bits, was employed on the basis of its excellent performance in comparisons of simulated natural products[38] and chemical similarity search more generally[39,40]. We note that notwithstanding the excellent performance of the ECFP6 fingerprint in these benchmarks, this algorithm tends to produce 'sparse' fingerprints (that is, bit vectors in which most bits are not set), and consequently will generally yield low Tcs for any comparison of two structures that are not perfectly identical[13]. To contrast the observed Tcs with random expectation, we therefore additionally calculated the Tc between PRISM 4 predicted structures and true secondary metabolite structures from all of the remaining, non-matching BGCs. For PRISM and NP.searcher, which can generate more than one predicted structure for a given BGC, the median Tc was compared; when a BGC was associated with more than one product, the median over all pairwise comparisons was calculated. A maximum of 100 predicted structures were considered for each BGC from PRISM and NP.searcher. We also assessed the distribution of functional groups in true and predicted structures, using the algorithm proposed by Ertl[14] and implemented in the RDKit to identify functional groups in an unbiased manner, without relying on a prespecified list of manually curated functional groups. For this analysis, one predicted or true structure was randomly selected for each BGC. NP.searcher[10] source code was obtained from the web server at http://dna.sherman.lsi.umich.edu/ and run with the default mass window parameters (1–5,000 Da), and all of cyclization, glycosylation, and dimerization enabled. antiSMASH 5.1.2 (ref. [7]) was obtained from Bioconda[41], and run with default settings. Sites of variability or uncertainty, denoted in SMILES output by antiSMASH as [Rn], where n is an integer, and in SMILES output by NP.searcher as [X], were replaced with the wild card symbol [*] in order to parse predicted structures with the RDKit. Statistical significance was assessed using the Brunner–Munzel paired rank test[42], a nonparametric test of the difference in medians robust to differences in the shape of the distributions being compared[43], except in Fig. 2e, where the $t$-test was used instead because the Brunner–Munzel test produces inflated $p$-values in comparisons involving fewer than 10 observations[44].

**Analysis of secondary metabolism in 10,121 prokaryotic genomes**. We used PRISM 4 and antiSMASH 5 to predict the chemical structures of secondary metabolites encoded within 3759 complete bacterial genomes and 6362 metagenome-assembled genomes (MAGs). All bacterial genomes with an assembly level of 'Complete' were downloaded from NCBI Genome, and a set of dereplicated genomes as determined by the Genome Taxonomy Database[15] were retained to mitigate the impact of highly similar genomes on our analysis. Similarly, a set of 7902 MAGs[23] was obtained from NCBI BioProject (accession PRJNA348753) and the subset of dereplicated genomes was retained. Detected BGCs were matched between PRISM and antiSMASH if their nucleotide sequence overlapped over any range. A small number of PRISM BGC types were mapped to more than one antiSMASH BGC type, including aminoglycosides ('amglyccycl' and 'oligo-saccharide'), type I polyketides ('t1pks' and 'transatpks'), and RiPPs ('bottromycin', 'cyanobactin', 'glycocin', 'head_to_tail', 'LAP', 'lantipeptide', 'lassopeptide', 'linar-idin', 'microviridin', 'proteusin', 'sactipeptide', and 'thiopeptide'). The "hybrid" category encompassed all BGCs assigned any combination of two or more cluster types, i.e., it was not limited to hybrid NRPS-PKS BGCs. The "other" category encompassed aryl polyenes, bacteriocins, butyrolactones, ectoines, furans, homo-serine lactones, ladderanes, melanins, N-acyl amino acids, NRPS-independent siderophores, phenazines, phosphoglycolipids, resorcinols, stilbenes, terpenes, and

type III polyketides. Producing organism taxonomy was based on genome phylogeny and retrieved from the Genome Taxonomy Database[15].

Cheminformatic metrics, including molecular weight, number of hydrogen bond donors and acceptors, octanol-water partition coefficients, and Bertz topological complexity, were calculated in RDKit. Both platforms occasionally generated very small, non-specific structure predictions (for example, a single unspecified amino acid or a single malonyl unit) that did not provide actionable information about the chemical structure of the encoded product; to remove these from consideration, we applied a molecular weight filter to remove structures under 100 Da output by either platform. To evaluate the internal structural diversity of each set of predicted structures, we computed the distribution of pairwise Tcs for each set[45], taking the median pairwise Tc instead of the mean as a summary statistic to ensure robustness against outliers. Structural similarity to known natural products was assessed using the RDKit implementation of the 'natural product-likeness' score[22], and by the median Tc between predicted structures and the known secondary metabolite structures deposited in the NP Atlas database[46].

**Quantitative predicted structure-activity relationship modeling of encoded secondary metabolites**. To evaluate the possibility of computationally inferring the likely activity of an encoded secondary metabolite based on its predicted chemical structure in PRISM 4, we undertook an extensive literature review to assign antibacterial, antitumor, immunomodulatory, antifungal, and/or antiviral activities to the metabolites within our 'gold standard' set of BGCs. In total, 833 of 1281 BGC products were assigned to at least one of the five activity classes. SVMs were trained in Python using the 'scikit-learn' package, using the hyperparameters recommended by Olson et al.[47]. Classifiers trained on 1024-bit hashed ECFP6 chemical fingerprints of PRISM predicted structures were compared to classifiers trained on counts of Pfam domains[48], identified using Pfam version 31.0 and HMMER version 3.1b2. When more than one structure was predicted for a given BGC, the value of each bit was averaged over all predicted structures (thus, for instance, if a given bit had a value of 1 in two of ten predicted structures for a given cluster, the value of that feature was set to 0.2). Accuracy was assessed using tenfold cross-validation. The statistical significance of classifier performance, as quantified by the area under the ROC curve (AUC), was first evaluated relative to random expectation using the Wilcoxon rank–sum test[24]. The AUC of classifiers trained on PRISM predicted structures was subsequently compared to those trained on Pfam domains using one-sided DeLong tests[49], which were combined by meta-analysis using Fisher's method.

For QPSAR modeling of predicted structures encoded in the complete collection of 10,121 bacterial genomes, a probability threshold corresponding to a 10% FDR was calculated from the ROC curves of each activity classifier based on tenfold cross-validation. Classifiers were subsequently trained on the entire 'gold standard' set and used to predict biological activity for genomically encoded BGC products, averaging variable bits across predicted structures as above. Visualization of the complete chemical space of predicted structures was accomplished using UMAP[25], using the implementation in the 'uwot' R package, and the first three principal components of the chemical fingerprint matrix as input. The 10 nearest neighbors were used for manifold approximation, with all other parameters set to their default values. A single predicted chemical fingerprint was sampled at random from each BGC in cases where more than one existed, and duplicate fingerprints were removed. To mitigate overplotting, 50% of points in the UMAP manifold were sampled at random for plotting.

**Software**. Statistical analyses were performed in R, using the 'nparcomp', 'AUC', 'pROC' and packages. Other aspects of data analysis were performed with the 'jsonlite', 'magrittr', 'tidyverse', and 'uwot' packages. Plotting was performed with the 'ggplot2' and 'patchwork' packages. See the Life Sciences Reporting Summary for further details.

**Reporting summary**. Further information on research design is available in the Nature Research Reporting Summary linked to this article.

## Data availability
The genomes analyzed in this study are publicly available from the NCBI Genome database and the Sequence Read Archive (accession PRJNA348753). Predicted and true chemical structures from the 'gold standard' set of 1,281 BGCs are provided in Supplementary Data 2. Predicted chemical structures from the collection of 10,121 complete or metagenome-assembled prokaryotic genomes analyzed in this study are provided in Supplementary Data 3. FASTA files for the 'gold standard' BGCs are available via Zenodo (https://doi.org/10.5281/zenodo.3985982). PRISM output files, in JSON format, for all of the genomes analyzed in this study are available via Zenodo (https://doi.org/10.5281/zenodo.3985978). Source data are provided with this paper. Source data are provided with this paper.

## Code availability
Source code used to conduct the analyses described in the manuscript is available from GitHub (https://github.com/Adapsyn/prism-4-paper).

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

## Acknowledgements

We thank D. Capstick, M. Chasse, K. Dial, W. Mousa, and L. Zettle for assistance with curation of newly published BGCs, and B. Lake, D. Levin, Y. Lin, V. Marando, and J. Pierscianowski for assistance with chemical structure curation. This work was supported by a Canada Research Chair (CIHR) New Investigator Award (to N.A.M.); an Ontario Early Investigator Award (to N.A.M.); the Canada Research Chairs Program (to N.A.M.); a CIHR Operating Grant (to N.A.M.); and a CIHR–Joint Programming Initiative on Antimicrobial Resistance Research Grant (to N.A.M.).

## Author contributions

M.A.S. developed PRISM 4, performed analysis, and wrote the manuscript. C.W.J. contributed to prediction of most secondary metabolite classes. M.G. contributed to expanded thiotemplated prediction. C.A.D. and N.J.M. contributed to the analysis. R.J.M. organized curation of the 'gold standard' BGC set. A.M.K., M.P.T.C., A.P., N.S., and Q.H.T., performed curation of 'gold standard' BGCs and their products. M.R.M.R. and H.L. contributed to aminoglycoside prediction. A.L.H.W. contributed to β-lactam prediction. D.P.W. contributed to data analysis and web server development. N.A.M. supervised the project.

## Competing interests

N.A.M. is a founder of Adapsyn Bioscience. M.A.S. and C.W.J. are or were at one time consultants to Adapsyn Bioscience. M.G., N.J.M., A.M.K., R.J.M., H.L., A.P., N.S., D.P.W., and C.A.D. are or were at one time employed by Adapsyn Bioscience. The remaining authors declare no competing interests.
