## [Peer Review File · Nature Communications]

REVIEWER COMMENTS

Reviewer #1 (Remarks to the Author):

In this manuscript, Magarvey and colleagues report on PRISM 4, a platform for the prediction of chemical structures for genomically encoded antibiotics from a variety of families (e.g., NRPs, PIs, aminoglycosides, beta-lactams, RIPPs, etc). This optimized program presents some advantages over PRISM, antiSMASH, and NP.Predictor. The manuscript is also well written. It is important to note that the Supplementary Figures and Tables were not provided.

Besides this omission, there is one major issue that needs to be addressed for this manuscript to be of the caliber of the journal:

1. In order to confirm the impact of this program, the authors should identify at least 2 new structures (not NRPs or PKs as these could be identified by previous version of PRISM) (antibiotics or compounds with other biological activities), isolate the compounds and confirm the structures of these compounds via traditional biophysical methods. This is essential to show that the new structures (which are the ones that would be interesting. NOTE: known compounds are not exciting as they are already known.) with unusual scaffolds can be predicted correctly. Without these confirmations of new structures, the article does not raise to the impact of this journal. This should be pretty simple to do and will bring the manuscript to an appropriate level. Without this, the manuscript offers only some computational advantage of predicting (without real confirmation) higher percentage of structures than other programs.

Reviewer #2 (Remarks to the Author):

The authors present here PRISM 4, a software tool able to take genomic sequences as input and predict the chemical structures of the pathways they encode, including antibiotics. This type of tool is of great utility for synthetic biology efforts in general, and to tackle the need for novel antibiotics in particular.

The authors showcase the capabilities of PRISM 4 in four different ways. First, they compare its efficiency to antiSMASH and NP.searcher by creating and using a gold standard data set comprising 1,281 manually curated biosynthetic gene clusters (BGC). Second, they test its performance vs antiSMASH using a collection of 3,759 complete bacterial genomes. Third, they compare the results of PRISM 4 vs antiSMASH by using 6,362 genomes obtained from metagenomes, since these are the main source of metabolically diverse sequence nowadays. Fourth, they use the predicted structure of the metabolite as input for SVMs that predict the probability that a BGC encodes a metabolite that has antibacterial, antifungal, antiviral, antitumor, or immunomodulatory activity (so as to prioritize further study).

In a majority of cases, PRISM 4 is shown to compare favorably to the few competitors available. PRISM 4 represents, hence, the latest and most comprehensive effort in translating sequences to the antibiotics they encode. For this reason, I think PRISM 4 is a welcome addition to literature. However, before considering publication a few major issues should be solved:

A)The gold data set comprising manually curated 1,281 BGCs must be shared. In terms of computational biology, curated data sets such as this one are as valuable as new tools. This gold data set is the most comprehensive to date and will be incredibly valuable to the developers of other algorithms in the future. Also, it is fundamental in order to reproduce the results in the paper. This reviewer has seen the results of comparisons in Supplementary Table 3, but has not been able to find the corresponding fasta files. It should not be this complicated. Also, the file name "249991_0_data_0qc9I8c" is anything but helpful and intuitive. <ED: unfortunately the manuscript submission system renames files in non-intuitive ways, so we know this was not your doing!>.

B)The code must be made available, in order to enable reproducing the results. We understand that this code is the basis of a company, but dual licenses can be created for research and commercial use. Since this is a scientific journal and the hallmark of science is reproducibility, publication in this journal must allow for reproducing the results obtained. Hence the need for the code and the data sets.

Other minor issues that need to be addressed in order to facilitate understanding to the general reader of a multidisciplinary journal such as Nat. Comm. are:

- 1) Please use line numbers to facilitate the job of reviewers.
- 2) Please provide a more thorough quantitative comparison with PRISM 3: how many classes are newly covered? How many more reactions were added? Please justify that PRISM 4 is not a minor incremental advance from PRISM 3.
- 3) HMMs were used to annotate microbial genome sequences, whereas the PI has advertised his use of deep learning (DL) in several talks. Please comment on the use of DL, and its advantages/disadvantages with respect to HMMs.
- 4) The phrase starting as "Here, we present PRISM 4, which enables genome-guided..." in page 2 is unduly complicated. Please rephrase.
- 5) Devote 1-2 sentences to provide an intuitive explanation of the Tanimoto coefficient to the casual reader. Provide examples so the reader can get an intuitive idea of the scale: is 0.5 good enough, or should we aim for 0.8?
- 6) This reviewer does not appreciate "a high degree of predictive accuracy across a wide range of secondary metabolite classes" (page 3) in figure 2e. Tanimoto coefficients of 0.25 for aminocoumarin and type 1 and 2 polyketide do not represent a high degree of predictive accuracy. Please rephrase to comment on the differences for each family or provide a comparison that justifies the "high degree of predictive accuracy".
- 7) Provide a one or two sentence description of the Jensen-Shannon divergence in 2d, for the non expert reader.
- 8) The comparison to NP.searcher in Fig. 2 is dropped for Fig. 3 without comment. Please justify explicitly.
- 9) Page 5: in phrase "several bacterial phyla whose biosynthetic capacity has historically not been widely appreciated" please provide some examples.
- 10) Devote 1-2 sentences to provide an intuitive explanation of the "rule of five" for the non-expert reader.
- 11) Devote 1-2 sentences to provide an intuitive explanation of "Bertz's topological complexity index" for the non-expert reader.
- 12) Page 6: "these models yielded significantly more accurate predictions of biological activity than random expectation". It is now clear how this was determined. Please explain.
- 13) Devote 1-2 sentences to provide an intuitive explanation of "UMAP" for the non-expert reader.
- 14) This reviewer has a hard time understanding the utility of the UMAP plots 4c and 4d. Please further explain the "take home message" and why it is important.

Reviewer #3 (Remarks to the Author):

This paper presents the fourth version of the genome mining tool PRISM and attempts to improve the secondary metabolite structure prediction using the biosynthetic gene cluster sequences found in bacterial genomes and metagenomes.

The platform can be used via an interactive webpage and works well. The updated version includes chemical structure prediction for 16 different classes of secondary metabolites via an increased amount of HMMs and implemented tailoring reactions.

The webpage layout is simple, tidy and easy to use. Also, it's satisfying to be able to select further advanced options for an optimized search. Overall, this is a nice tool and worth publishing, if the

improvements stated in the manuscript can somehow be validated. The authors compare their work mostly to the most commonly used tool AntiSMASH. Figure 3 shows a large overlap of predictions with both methods, and a number of unique clusters that are only predicted with PRISM4. However, it is not clear to me, if these are true BGCs. There is no validation. In addition, some of the HMM cutoffs are very low (supplementary excel sheet: 2a). The cutoffs for most of the RiPPs or bacteriocin HMMs, for example, are on average much lower. Interestingly, these are the classes that PRISM4 exceeds over antiSMASH and raises questions for their accurate predictions.

Further comments:

Comparisons of PRISM4 with antiSMASH4 although seem to show prediction improvements, are not timely as antiSMASH5 has already been around for 1.5 years. High scoring PRISM4 predictions may be compared with antiSMASH5 predictions.

The source code for PRISM4 is not accessible via the link mentioned in the manuscript.

Many academic journals have a policy in place for making the scientific software available to users without making them register on such servers. The current implementation of PRISM4 mandates acceptance of NFP EULA and registration of academic users. This might restrict many researchers in using this tool, and overall community acceptance of these novel methods. The editorial policy of N.COMM in this aspect is not clear.

Figure2: Information in plot c and e should ideally tally. The highest median Tanimoto coeff. in plot c is less than 0.6, while for at least 7 BGC families this value is greater than 0.6.

GUI had issues with correctly displaying the predicted structures when tested on both Chrome and Safari web-browser.

Some bugs in correctly displaying the colours for a particular BGC type were observed.

Limitations of the structure prediction method may be discussed in more details.

Figure3b and 4b depicts Firmicutes, Firmicutes_A, Firmicutes_B and Firmicutes_C phylum. These phyla names are according to which taxonomy, it is not clear.

Response to reviewers

We would like to thank the editor and the three reviewers for their thoughtful and constructive comments on the manuscript. We were grateful for their enthusiasm for our work, and have taken the opportunity to revise our manuscript in order to provide comprehensive responses to all points raised during the review process. We have highlighted all changes to the text in the accompanying manuscript and include new figure panels and text inline in the following responses for convenience. The most important changes are as follows:

- We have updated all of the analyses in **Figures 2 and 3**, and the corresponding Supplementary Figures, to use antiSMASH 5, the latest release of the antiSMASH platform.
- To validate the BGC detection functionality of PRISM 4, we carried out a blinded investigation of a large random sample of BGCs uniquely detected only by either PRISM or antiSMASH, finding a significantly lower false-positive rate among PRISM-only BGCs.
- We provide the complete set of FASTA files underlying our gold standard benchmark dataset via Zenodo as a benchmarking resource for the field.
- We provide the PRISM output files, in JSON format, for all of the genomes analyzed in this study, as well as the source code used to analyze these files, in order to ensure the reproducibility of our results.
- We have revised the Results section to provide a more thorough exposition of our analytical choices (e.g., providing more detailed explanations of key metrics) for the general reader.
- New text has been added to the Discussion, commenting on the limitations of PRISM 4 and future directions for BGC structure prediction.
- Finally, we enclose the PRISM 4 source code with this resubmission for confidential peer review.

Throughout this document, editor and reviewer comments are shown here in **blue**, with our own response in **black**, and changes to manuscript text or figure captions shown in **red**.

Reviewer #1 (Remarks to the Author):

In this manuscript, Magarvey and colleagues report on PRISM 4, a platform for the prediction of chemical structures for genomically encoded antibiotics from a variety of families (e.g., NRPs, PIs, aminoglycosides, beta-lactams, RiPPs, etc). This optimized program presents some advantages over PRISM, antiSMASH, and NP.Predictor. The manuscript is also well written. It is important to note that the Supplementary Figures and Tables were not provided.

Besides this omission, there is one major issue that needs to be addressed for this manuscript to be of the caliber of the journal:

1. In order to confirm the impact of this program, the authors should identify at least 2 new structures (not NRPs or PKs as these could be identified by previous version of PRISM) (antibiotics or compounds with other biological activities), isolate the compounds and confirm the structures of these compounds via traditional biophysical methods. This is essential to show that the new structures (which are the ones that would be interesting. NOTE: known compounds are not exciting as they are already known.) with unusual scaffolds can be predicted correctly. Without these confirmations of new structures, the article does not raise to the impact of this journal. This should be pretty simple to do and will bring the manuscript to an appropriate level. Without this, the manuscript offers only some

computational advantage of predicting (without real confirmation) higher percentage of structures than other programs.

We thank the reviewer for their careful evaluation of our work, and sincerely apologize for the error in providing the Supplementary Figures and Tables. The reviewer is, of course, correct to point out that the predicted structures described in this manuscript have not been experimentally verified. However, our intention was not to assert that these represent unambiguously correct structure predictions. Rather, we aimed to establish the value of PRISM 4 as a resource for genome mining and the investigation of secondary metabolism more generally. In this regard, we feel our large-scale benchmark, achieved through the extensive curation of a comprehensive set of 1,281 BGCs with known products, provides fundamentally a more informative assessment, as it provides a truly comprehensive outlook on structure prediction accuracy as opposed to one or two novel case studies. Moreover, we have demonstrated the value of PRISM 4 for the interrogation of secondary metabolism in large collections of reference genomes or metagenome-assembled genomes, and shown that these predicted structures enable more accurate inference of biological activity. Not only does this define a “roadmap” for the targeted discovery of novel, biologically active agents, but it also enables investigators to address systems-level questions that could not have been answered before—for instance, our finding that human-associated microbes are significantly more likely to produce immunomodulatory compounds than their environmental counterparts. Thus, in summary, we feel that PRISM 4 represents a significant advance that will widely benefit the community.

Reviewer #2 (Remarks to the Author):

The authors present here PRISM 4, a software tool able to take genomic sequences as input and predict the chemical structures of the pathways they encode, including antibiotics. This type of tool is of great utility for synthetic biology efforts in general, and to tackle the need for novel antibiotics in particular.

The authors showcase the capabilities of PRISM 4 in four different ways. First, they compare its efficiency to antiSMASH and NP.searcher by creating and using a gold standard data set comprising 1,281 manually curated biosynthetic gene clusters (BGC). Second, they test its performance vs antiSMASH using a collection of 3,759 complete bacterial genomes. Third, they compare the results of PRISM 4 vs antiSMASH by using 6,362 genomes obtained from metagenomes, since these are the main source of metabolically diverse sequence nowadays. Fourth, they use the predicted structure of the metabolite as input for SVMs that predict the probability that a BGC encodes a metabolite that has antibacterial, antifungal, antiviral, antitumor, or immunomodulatory activity (so as to prioritize further study).

In a majority of cases, PRISM 4 is shown to compare favorably to the few competitors available. PRISM 4 represents, hence, the latest and most comprehensive effort in translating sequences to the antibiotics they encode. For this reason, I think PRISM 4 is a welcome addition to literature. However, before considering publication a few major issues should be solved:

We thank the reviewer for their positive and thoughtful comments on our work.

A)The gold data set comprising manually curated 1,281 BGCs must be shared. In terms of computational biology, curated data sets such as this one are as valuable as new tools. This gold data set is the most comprehensive to date and will be incredibly valuable to the developers of other algorithms in the future. Also, it is fundamental in order to reproduce the results in the paper. This reviewer has seen the results of comparisons in Supplementary Table 3, but has not been able to find

the corresponding fasta files. It should not be this complicated. Also, the file name “249991_0_data_0qc9l8c” is anything but helpful and intuitive. <ED: unfortunately the manuscript submission system renames files in non-intuitive ways, so we know this was not your doing!>.

We are grateful to see the reviewer recognize the utility of our extensively curated, ‘gold standard’ resource of BGC sequences and the structures of their corresponding products. Indeed, as described in our manuscript, a tremendous amount of effort went into the curation of this resource. As such, we were very happy to see the reviewer describe this resource as “incredibly valuable” to future developers and are happy to make the FASTA files publicly available, complementing the SMILES of the structures as provided with the initial manuscript. We now provide all 1,281 FASTA files through a Zenodo deposition, in order to associate this resource with its own unique DOI.

This resource is noted in the revised Data Availability statement (lines 510-517):

Data availability. The genomes analyzed in this study are publicly available from the NCBI Genome database and the Sequence Read Archive (accession PRJNA348753). Predicted and true chemical structures from the ‘gold standard’ set of 1,281 BGCs are provided in **Supplementary Table 3**. Predicted chemical structures from the collection of 10,121 complete or metagenome-assembled prokaryotic genomes analyzed in this study are provided in **Supplementary Table 4**. **FASTA files for the ‘gold standard’ BGCs are available via Zenodo (<https://doi.org/10.5281/zenodo.3985982>). PRISM output files, in JSON format, for all of the genomes analyzed in this study are available via Zenodo (<https://doi.org/10.5281/zenodo.3985978>).**

Finally, we apologize for the non-intuitive file naming; as noted by the editor this is imposed by the manuscript submission system, and was not our intention.

B)The code must be made available, in order to enable reproducing the results. We understand that this code is the basis of a company, but dual licenses can be created for research and commercial use. Since this is a scientific journal and the hallmark of science is reproducibility, publication in this journal must allow for reproducing the results obtained. Hence the need for the code and the data sets.

We appreciate that the reviewer recognizes that the source code underlying PRISM 4 “forms the basis for a company.” We would go a step further: this unique technology is central to Adapsyn’s commercial viability. For this reason, we regret that it simply is not possible to provide this code to the general public. We believe that the provision of PRISM 4 to the community as a freely available resource gives users ample ability to test and reproduce the main claims of our manuscript, as well as to analyze their own genomes. As such, we feel the PRISM 4 web server provides a tremendously useful resource to the community. We also want to clarify that this service carries no cost to users: Adapsyn bears the expense of all necessary computing resources.

We are, however, sensitive to the issue of reproducibility and agree with the reviewer’s broader point about the reproducibility of the results. To this end, we have taken three additional steps:

1. We provide all of the PRISM output files, in JSON format, for all of the genomes analyzed in this study, via a second Zenodo deposition. These files encode all of the information used by PRISM internally to analyze genomes for BGCs and generate structure predictions, including the positions and identities of all biosynthetic domains, their links to tailoring reactions, and the complete biosynthetic pathway predictions for every detected BGC. More importantly, these files contain all of the information used to conduct the analyses reported in the

manuscript. Consequently, we feel this addition alone provides a decisive step towards reproducibility.

2. In addition to the data files themselves, we additionally provide the source code used to conduct the analyses described in this manuscript, which is available from GitHub (<https://github.com/Adapsyn/prism-4-paper>). In conjunction with the underlying data files, these two steps ensure the reproducibility of our results, addressing the reviewer's central concern.
3. Finally, we are providing the PRISM 4 source code itself for *confidential* peer review, to allow its assessment only by the reviewers. This code is included with the resubmitted manuscript, with the understanding it will not accompany publication.

Collectively, we believe these steps go above and beyond current standards for reproducibility in the field, and will ensure that interested readers are able to reproduce our entire analysis. These changes are noted in the revised Data Availability and Code Availability statements (lines 510-521):

Data availability. The genomes analyzed in this study are publicly available from the NCBI Genome database and the Sequence Read Archive (accession PRJNA348753). Predicted and true chemical structures from the 'gold standard' set of 1,281 BGCs are provided in **Supplementary Table 3**. Predicted chemical structures from the collection of 10,121 complete or metagenome-assembled prokaryotic genomes analyzed in this study are provided in **Supplementary Table 4**. **FASTA files for the 'gold standard' BGCs are available via Zenodo (<https://doi.org/10.5281/zenodo.3985982>).** **PRISM output files, in JSON format, for all of the genomes analyzed in this study are available via Zenodo (<https://doi.org/10.5281/zenodo.3985978>).**

Code availability. ~~The algorithm described in this paper is available from <https://grid.adapsyn.com/prism>.~~ **Source code used to conduct the analyses described in the manuscript is available from GitHub (<https://github.com/Adapsyn/prism-4-paper>).**

Other minor issues that need to be addressed in order to facilitate understanding to the general reader of a multidisciplinary journal such as Nat. Comm. are:

- 1) Please use line numbers to facilitate the job of reviewers.

We apologize for the inconvenience. We have added line numbers to the revised manuscript, and note the line numbers for changes made in the revised manuscript throughout this response.

- 2) Please provide a more thorough quantitative comparison with PRISM 3: how many classes are newly covered? How many more reactions were added? Please justify that PRISM 4 is not a minor incremental advance from PRISM 3.

We thank the reviewer for this excellent suggestion, which we have addressed by including a new supplementary figure depicting the number of biosynthetic domains and virtual tailoring reactions incorporated in PRISM 3 vs. PRISM 4. This figure, which is included below, reflects the addition of 1,083 new HMMs (an increase of 145%) as well as 334 reactions (an increase of 118%) since the release of PRISM 3—more than doubling the scope of PRISM 4.

Supplementary Fig. 10 | Comparison of PRISM 3 and PRISM 4. Number of biosynthetic domains, left, and virtual tailoring reactions, right, incorporated in PRISM 4 and the previous release, PRISM 3.

We discuss this figure, and these increases, in the revised Methods section (lines 371-373):

[...] The complete sets of HMMs and virtual tailoring reactions are enumerated in **Supplementary Table 2**, and further detail on these new classes is provided in the **Supplementary Note**. Compared to PRISM 3, PRISM 4 includes a total of 1,083 newly developed HMMs (an increase of 145%) and 334 new reactions (an increase of 118%) that were not included in previous versions (**Supplementary Fig. 10**).

3) HMMs were used to annotate microbial genome sequences, whereas the PI has advertised his use of deep learning (DL) in several talks. Please comment on the use of DL, and its advantages/disadvantages with respect to HMMs.

We have added a new paragraph to the Discussion commenting on recent deep learning-based approaches to BGC detection and annotation by us (Merwin et al., *PNAS* 2020, doi: 10.1073/pnas.1901493116) and others (Hannigan et al., *Nucleic Acids Res.* 2019, doi: 10.1093/nar/gkz654) to address this request. In addition, based on feedback from reviewer #3, we also provide more extensive discussion of the limitations of our approach. The newly added paragraph (lines 292-309) is reproduced below:

Some limitations should be noted. In developing PRISM 4, we set out to codify an enormous corpus of knowledge, accumulated over decades of research in biosynthesis and enzymology, into an algorithmically tractable form. An inevitable consequence of this approach is that PRISM relies on homology between newly detected proteins and known enzymatic machinery in order to reveal BGCs and predict the structures of their genetically encoded products. For this reason, PRISM can neither identify BGCs from undescribed families, nor predict novel enzymatic activities. More generally, current models of secondary metabolite biosynthesis are incomplete, which places an inherent limit on the accuracy of structure prediction; we have sought to address this by revising the systems used for BGC detection and structure prediction as new information has become available. Recently, we and others have shown that deep learning-based methods can enable more flexible and accurate detection or characterization of BGCs or individual biosynthetic components^{27,28}. However, at present these approaches still rely on interfacing with rule-based systems such as that employed by PRISM 4 to permit structure prediction²⁷, or else are not capable of generating predicted structures²⁸. In the future, more sophisticated

machine-learning approaches might enable the end-to-end prediction of encoded small molecules directly from primary sequence. Finally, PRISM 4 was designed primarily for prokaryotic genome analysis and thus cannot identify BGCs families thought to be specific to eukaryotes, and—like all tools for genome annotation—may produce incongruous results when applied to fragmented or low-quality genome assemblies.

4) The phrase starting as “Here, we present PRISM 4, which enables genome-guided...” in page 2 is unduly complicated. Please rephrase.

We have rephrased this sentence as follows (line 64 of the revised manuscript):

Here, we present PRISM 4, which enables genome-guided chemical structure prediction for every class of bacterial natural antibiotics currently in clinical use, including aminoglycosides, nucleosides, β -lactams, alkaloids, and lincosamides among other entirely new classes of metabolites. Moreover, PRISM 4 achieves a dramatic increase in coverage of enzymatic tailoring reactions encoded within canonical thiotemplated pathways (**Fig. 1** and Methods).

5) Devote 1-2 sentences to provide an intuitive explanation of the Tanimoto coefficient to the casual reader. Provide examples so the reader can get an intuitive idea of the scale: is 0.5 good enough, or should we aim for 0.8?

7) Provide a one or two sentence description of the Jensen-Shannon divergence in 2d, for the non expert reader.

10) Devote 1-2 sentences to provide an intuitive explanation of the “rule of five” for the non-expert reader.

11) Devote 1-2 sentences to provide an intuitive explanation of “Bertz’s topological complexity index” for the non-expert reader.

(Note that we have slightly reorganized these comments in order to address these related points.) We agree with the reviewer’s broad point that the presentation of these analyses would benefit from a more complete exposition. We have revised this section to include brief descriptions of the Tanimoto coefficient, the Jensen-Shannon divergence between functional group distributions, Lipinski’s rule of five, and the Bertz topological complexity index, as requested. We hope this revised text clarifies these metrics. The edits made in the revised manuscript are as follows:

lines 94-100:

To quantify the similarity of predicted structures to the true cluster products, we calculated the Tanimoto coefficient¹³ (Tc) between real and predicted structures from each cluster, a measure of chemical similarity that reflects the fraction of substructures shared between the two molecules [...]. Using this metric, we found PRISM 4 achieved statistically significant predictive accuracy across a wide range of secondary metabolite classes (**Fig. 2e**).

lines 105-110:

We additionally quantified the accuracy of structure predictions based on the functional groups they contained¹⁴. Using the Jensen-Shannon divergence to compare the distributions of functional groups found in true and predicted structures, we observed that the functional group content of PRISM 4 predicted structures was significantly more similar to that of true products than that of structures predicted by antiSMASH or NP.searcher (bootstrap $p < 0.001$; **Fig. 2d**).

lines 170-183:

Because the true structures of the metabolites encoded by these loci are not known, we were unable to directly assess the accuracy of structure prediction. Instead, we asked whether predicted structures had

structural features characteristic of known natural products¹⁹. Previous studies have found that a relatively small proportion of natural products are within the chemical space defined by Lipinski's "rule of five," a set of guidelines developed to facilitate the design of orally bioavailable drugs^{20,21}. Relative to structures predicted by antiSMASH, a lower proportion of PRISM 4 predictions were within Lipinski's rule of five space¹⁷ (χ^2 test, $p < 10^{-15}$; **Fig. 3c**), having greater molecular weights (paired Brunner–Munzel test, $p < 10^{-15}$; **Fig. 3d**), more hydrogen bond donors and acceptors ($p < 10^{-15}$; **Supplementary Fig. 7a–b**), and greater octanol-water partition coefficients ($p < 10^{-15}$; **Supplementary Fig. 7c**). PRISM 4 predictions were also more structurally complex, as quantified using the Bertz topological complexity index¹⁸ ($p < 10^{-15}$; **Fig. 3e**), a measure of molecular complexity that incorporates both the complexity of the bonding and the distribution of heteroatoms. Moreover, PRISM 4 predictions were also more structurally diverse, as quantified by the median intra-set Tanimoto coefficient ($p < 10^{-15}$; **Fig. 3f**).

The reviewer's request for clarification regarding the meaning of the magnitude of the Tanimoto coefficient (Tc) brings up a more general point that is also relevant to the reviewer's following comment, and which we therefore feel merits discussion in slightly more detail. The Tc is defined for a pair of molecular 'fingerprints,' where a fingerprint is a binary vector that denotes the presence or absence of a series of chemical substructures in a molecule of interest. Specifically, the Tc is computed as the size of the intersection between the fingerprints (that is, the number of substructures shared between both molecules) divided by the union (that is, the number of substructures present in either molecule). A natural consequence of this definition is that the value of the Tc is highly sensitive to the set of substructures considered in the calculation.

In this manuscript, we use the ECFP molecular fingerprint, which enumerates all possible substructures within a given radius of each atom in the molecule (here, the radius is set to 3, producing fingerprints known as 'ECFP6' for their diameter). We have previously shown (Skinnider et al., *J. Cheminform.* 2017, doi: 10.1186/s13321-017-0234-y) that these fingerprints produce *relative rankings* of chemical similarity that are strongly aligned with the biosynthetic origins of natural products. Similar findings with respect to performance have consistently been reported in the context of virtual screening/quantitative structure-activity relationships (e.g., Riniker and Landrum, *J. Cheminform.* 2013, doi: 10.1186/1758-2946-5-26; O'Boyle and Sayle, *J. Cheminform.* 2016, doi: 10.1186/s13321-016-0148-0). However, these descriptors are well-known to produce 'sparse' fingerprints (that is, most bits are not set to 1), with the consequence that the *absolute values* of the ECFP6 Tc are generally low and not uniformly distributed in the range [0,1]. Moreover, different molecules—or even molecular pairs—will tend to produce characteristic Tc distributions on the basis of their relative sizes, which in turn roughly determine the magnitude of the denominator in the Tc calculation (i.e., the number of substructures present in both molecules). The implication is that for a Tanimoto coefficient of 0.5 may be "good enough" for some molecules, but not others. Finally, it is important to note that 'denser' fingerprints which may have a lower discriminatory power, such as the widely used MACCS keys, will reliably produce much higher absolute values, yet these will correspond less well to other indices of chemical similarity (e.g., shared biosynthetic origin, activity against a common target, or subjective judgement by medicinal chemists) and so be less useful in practice.

For these reasons, we are hesitant to provide a single "intuitive idea of the scale" as requested by the reviewer, at the risk of misleading the reader about the nature of this calculation. Instead we have tried to briefly allude to this complexity within the main text and discuss some of these points in the Methods. What we want to emphasize as the "take-home message" in this section is that PRISM 4 structure predictions are *more accurate*, in *relative* terms, than those from competing software packages (i.e., antiSMASH and NP.searcher).

6) This reviewer does not appreciate “a high degree of predictive accuracy across a wide range of secondary metabolite classes” (page 3) in figure 2e. Tanimoto coefficients of 0.25 for aminocoumarin and type 1 and 2 polyketide do not represent a high degree of predictive accuracy. Please rephrase to comment on the differences for each family or provide a comparison that justifies the “high degree of predictive accuracy”.

We thank the reviewer for raising this important comment. First of all, we hope the discussion above provides a measure of clarification on this point. However, to more directly help the reader develop some intuition about what would represent a ‘high degree’ vs. a ‘low degree’ of accuracy, we felt it would be useful to show the distribution of *random* Tanimoto coefficients for each biosynthetic class. To achieve this, we compared PRISM 4 predicted structures to the true structures of non-matching BGCs. We then calculated the median Tanimoto coefficient between PRISM predicted structures from each cluster, and all possible non-matching clusters. This effectively gives a null distribution of Tanimoto coefficients. We then plotted the distribution of random pairs alongside the median and maximum Tc, as originally shown in **Figure 2e**. Finally, we statistically compared the distribution of median Tcs to random Tcs, in all cases finding a statistically significant difference, and typically a highly significant one ($p < 10^{-4}$ for 12 of 15 families). The updated **Figure 2e** is shown below:

(see **Figure 2**) e, Median and maximum Tanimoto coefficients between true and predicted structures generated by PRISM 4 for the gold standard set, by biosynthetic family and compared to the median Tanimoto coefficient between predicted structures and non-matched BGCs (“random pairs”). Top, statistical significance of the comparison between median and random Tanimoto coefficients (***, $p < 0.001$; **, $p < 0.01$; *, $p < 0.05$, two-sided t-test). Bottom, number of BGCs from each family in the gold standard set. Box plots show median (horizontal line), interquartile range (hinges), and the smallest and largest values no more than 1.5 times the interquartile range (whiskers) throughout.

We have also revised the Results section to clarify this analysis (lines 94-100):

To quantify the similarity of predicted structures to the true cluster products, we calculated the Tanimoto coefficient¹³ between real and predicted structures from each cluster, a measure of chemical similarity that reflects the fraction of substructures shared between the two molecules, and compared these to predicted and true structures from random BGC pairs (Methods). Using this metric, we found PRISM 4 achieved statistically significant predictive accuracy across a wide range of secondary metabolite classes (**Fig. 2e**).

Finally, we have revised the Methods as follows (lines 417-427 and 441-445):

To quantitatively assess the accuracy of PRISM 4 chemical structure predictions with reference to these known products, we calculated the Tanimoto coefficient (Tc) between the chemical fingerprints for each pair of true and predicted structures¹³. The ECFP6 chemical fingerprint³⁴, with a length of 1,024 bits, was employed on the basis of its excellent performance in comparisons of simulated natural products³⁵ and chemical similarity search more generally^{36,37}. **We note that notwithstanding the excellent performance of the ECFP6 fingerprint in these benchmarks, this algorithm tends to produce ‘sparse’ fingerprints (that is, bit vectors in which most bits are not set), and consequently will generally yield low Tcs for any comparison of two structures that are not perfectly identical¹³. To contrast the observed Tcs with random expectation, we therefore additionally calculated the Tc between PRISM 4 predicted structures and true secondary metabolite structures from all of the remaining, non-matching BGCs. [...] Statistical significance was assessed using the Brunner–Munzel paired rank test⁴¹, a nonparametric test of the difference in medians robust to differences in the shape of the distributions being compared⁴², except in Figure 2e, where the t-test was used instead because the Brunner–Munzel test produces inflated p-values in comparisons involving fewer than 10 observations⁴³.**

8) The comparison to NP.searcher in Fig. 2 is dropped for Fig. 3 without comment. Please justify explicitly.

We dropped NP.searcher from further analysis on the basis that it was not designed as a comprehensive platform for BGC analysis, but only for the analysis of nonribosomal peptides and polyketides; as shown in **Figure 2a**, NP.searcher detects less than half of the clusters detected by PRISM or antiSMASH in the gold standard set. We have clarified this motivation, as requested, on lines 140-143 of the revised manuscript:

To gain a broader perspective on PRISM 4’s ability to predict encoded metabolite structures from genome sequence, we used PRISM 4 to analyze secondary metabolism in a collection of 3,759 dereplicated complete bacterial genomes¹⁵. **For this comparison, we focused on PRISM and antiSMASH, as platforms designed to analyze BGCs from a wide range of biosynthetic families.**

9) Page 5: in phrase “several bacterial phyla whose biosynthetic capacity has historically not been widely appreciated” please provide some examples.

We have revised the text to provide examples of three bacterial phyla which have been previously shown to possess biosynthetic capacity, but from which no natural products have been isolated to date. These examples highlight new potential directions for genome-guided discovery of cryptic natural products (lines 151-152):

PRISM 4 also generated the majority of structure predictions for several bacterial phyla whose biosynthetic capacity has historically not been widely appreciated, **such as Desulfobacterota, Spirochaetota, or Campylobacterota (Fig. 3b).**

12) Page 6: “these models yielded significantly more accurate predictions of biological activity than random expectation”. It is now clear how this was determined. Please explain.

We apologize that this point was not clear in the text. The calculation of statistical significance in question is derived from the area under the receiver operating characteristic curve (AUROC or AUC), which characterizes the trade-off between the true-positive and false-positive rates, as shown in **Figure 4a**. The statistical significance of this area, as compared to that expected for a random

predictor, can be computed based on the mathematical relationship between the AUC and the Mann-Whitney U statistic (Mason and Graham, *Q. J. R. Meteorol. Soc.* 2002, doi: 10.1256/003590002320603584). This is the analysis that was performed here. We have revised this section to clarify this point, and include a reference to the Mason and Graham reference within the main text (lines 228-230 of the revised manuscript):

We undertook an extensive literature review to systematically curate bioactivity data for the 1,281 BGCs in the gold standard set, and trained support vector machines (SVMs) to predict the probability that a given BGC produces a compound with antibacterial, antifungal, antiviral, antitumor, or immunomodulatory activity, using ten-fold cross-validation to evaluate model accuracy. **To evaluate the performance of these models, we calculated the area under the receiver operating characteristic curve (AUC), and compared the observed AUCs to those expected from random predictors²².** In all cases, these models yielded significantly more accurate predictions of biological activity than random expectation (all $p < 10^{-15}$, Wilcoxon rank-sum test; **Fig. 4a**).

- 13) Devote 1-2 sentences to provide an intuitive explanation of “UMAP” for the non-expert reader.
14) This reviewer has a hard time understanding the utility of the UMAP plots 4c and 4d. Please further explain the “take home message” and why it is important.

The analyses presented in this section of this manuscript describe the application of PRISM to predicted structures from 10,121 complete or metagenome-assembled prokaryotic genomes, producing an enormous wealth of data. We felt it would be desirable to provide a high-level visualization of the entire dataset to orient the user. However, the scale of the dataset precluded the application of many classic approaches that have been previously used in the field (e.g., similarity networks as in Cimermancic et al., *Cell* 2014, doi: 10.1016/j.cell.2014.06.034 or Doroghazi et al., *Nat. Chem. Biol.* 2014, doi: 10.1038/nchembio.1659). UMAP is a non-linear dimensionality reduction technique that has recently found widespread application in the field of single-cell genomics.

In the context of this manuscript, UMAP has two very desirable properties: (i) its approach, based on local connectivity within a high-dimensional manifold, is compatible with very sparse data—such as single-cell gene expression measurements, but also ECFP6 fingerprints; and (ii) unlike other non-linear dimensionality reduction methods such as t-SNE, UMAP approximately preserves global structure, meaning points that are close in the low-dimensional space are also close in the high-dimensional space, and vice-versa.

We have previously found success using UMAP to visualize the global distribution of RiPP chemical diversity (Merwin et al., *PNAS* 2020, doi: 10.1073/pnas.1901493116), and so we reasoned it would provide a similar high-level overview of the predicted chemical landscape here. As requested, we have revised the manuscript to justify the use of UMAP in the Results section (lines 258-265):

To obtain a global overview of the chemical diversity within this dataset, we applied the non-linear dimensionality reduction technique UMAP (uniform manifold approximation and projection)²⁵ to the chemical fingerprints of PRISM predicted structures. Unlike some other non-linear dimensionality reduction methods, UMAP approximately preserves global structure, meaning points that are close in the low-dimensional space are also close in the high-dimensional space, and vice-versa. This visualization of the complete predicted chemical space of PRISM predicted structures revealed substantial chemical diversity within each bioactivity class (Fig. 4c).

Reviewer #3 (Remarks to the Author):

This paper presents the fourth version of the genome mining tool PRISM and attempts to improve the secondary metabolite structure prediction using the biosynthetic gene cluster sequences found in bacterial genomes and metagenomes.

The platform can be used via an interactive webpage and works well. The updated version includes chemical structure prediction for 16 different classes of secondary metabolites via an increased amount of HMMs and implemented tailoring reactions.

The webpage layout is simple, tidy and easy to use. Also, it's satisfying to be able to select further advanced options for an optimized search. Overall, this is a nice tool and worth publishing, if the improvements stated in the manuscript can somehow be validated.

We thank the reviewer for their thoughtful and positive comments on our work.

The authors compare their work mostly to the most commonly used tool AntiSMASH. Figure 3 shows a large overlap of predictions with both methods, and a number of unique clusters that are only predicted with PRISM4. However, it is not clear to me, if these are true BGCs. There is no validation. In addition, some of the HMM cutoffs are very low (supplementary excel sheet: 2a). The cutoffs for most of the RiPPs or bacteriocin HMMs, for example, are on average much lower. Interestingly, these are the classes that PRISM4 exceeds over antiSMASH and raises questions for their accurate predictions.

The reviewer raises the possibility that some of the clusters uniquely detected by PRISM 4, but not antiSMASH, may represent false positives. In the past, we have performed detailed manual reviews of clusters detected only by PRISM vs. antiSMASH (e.g. Skinnider et al., *PNAS* 2016, doi: 10.1073/pnas.1609014113), which led us to conclude that many of the PRISM-only BGCs represent *bona fide* true positives, whereas a substantial proportion of antiSMASH-only BGCs likely represent spurious predictions.

To address this comment, we therefore performed a similar review of a random sample of 200 BGCs detected only by one platform or the other (that is, 100 PRISM-only BGCs and 100 antiSMASH-only BGCs) in the present dataset. A critical difference in this evaluation, relative to our previous effort, is that we undertook the present validation in a blinded manner: investigators were blinded to which program detected the BGC of interest when calling false-positives. In this blinded, manual review, 55 of 100 antiSMASH-only clusters were annotated as false-positives, as compared to 37 of 100 PRISM-only clusters, a statistically significant difference ($p = 0.016$, χ^2 test).

We have incorporated the results of this analysis in a new paragraph in the Results section, and a new supplementary figure. In the interest of fairness, we note to the reader that a trade-off between sensitivity and specificity is inherent in any prediction task, and antiSMASH's higher false-positive rate will expectantly also allow it to achieve a lower false-negative rate on divergent or novel BGC types. These additions are included below:

To evaluate the BGC detection functionality of PRISM and antiSMASH, we carried out a blinded review of 200 randomly sampled clusters detected only by one of the two methods. Manual annotation suggested up to 55% of antiSMASH-only BGCs represented false positives (FPs), compared to up to 37% of PRISM-only BGCs ($p = 0.016$, χ^2 test; **Supplementary Fig. 8**). Among antiSMASH-only BGCs, recurrent categories of FPs included minimal fatty acid synthases, DUF692-associated bacteriocins, putative phosphonate BGCs associated with cell wall biosynthesis machinery, and isolated prenyltransferases classified as terpene BGCs. It should be noted that a trade-off between specificity and sensitivity is inherent to any prediction task, and the higher rate of FPs for antiSMASH also expectantly affords it a greater ability to detect—though not to predict structures for—novel or divergent new BGC types.

Supplementary Fig. 8 | Blinded manual review of BGC detection in PRISM 4 and antiSMASH 5. The proportion of true-positive clusters among two sets of 100 BGCs detected only by antiSMASH 5 and 100 BGCs detected only by PRISM 4, respectively, are shown. Error bars show the standard deviation of the sample proportion.

Finally, to address the reviewer’s concern regarding the HMM cutoffs in Supplementary Table 2, we feel it is useful to clarify the relationship between the HMM score cutoff and the length of the alignment. The plot below depicts the relationship between alignment length, in nucleotides, and HMM score cutoff for each of the 1,772 HMMs in PRISM 4. These two values are closely connected, in that the length of the alignment effectively bounds the range of scores that is possible for a homologous sequence.

Response figure 3.1 | Relationship between multiple sequence alignment length, in nucleotides, and HMM score cutoff in PRISM 4.

A natural consequence of this relationship is that for very short amino acid sequences, a low cutoff may nonetheless represent a stringent criterion for homology detection. Indeed, both of the specific

classes the reviewer mentions (RiPPs and bacteriocins) are noted for the presence of very short 'precursor' peptides, which naturally require lower cutoffs. We feel that ultimately, the accuracy achieved by PRISM 4 in structure prediction across a range of biosynthetic families (**Figure 2**) substantiates our view that these cutoffs have been appropriately set, in a manner that is suitable for the length of each individual multiple sequence alignment.

Further comments:

Comparisons of PRISM4 with antiSMASH4 although seem to show prediction improvements, are not timely as antiSMASH5 has already been around for 1.5 years. High scoring PRISM4 predictions may be compared with antiSMASH5 predictions.

We thank the reviewer for this comment. At the time this research was completed, antiSMASH 4 was the state-of-the-art release of this widely used platform. However, we recognize that in the interim, a newer release of the antiSMASH platform has been published. To address this point, we have re-done all of the analyses in the manuscript using antiSMASH 5, instead of antiSMASH 4.

The changes in this version, relative to version 4 (described in Blin et al., *Nucleic Acids Res.* 2019, doi: 10.1093/nar/gkz310), primarily impact on functionality such as output format, code runtime, and interfacing with other tools (e.g., ARTS or BiG-SCAPE), with comparatively minor updates to cluster detection and structure prediction. Accordingly, we observed a consistent, but very modest, improvement over antiSMASH 4 in our previous results. The net effect is that none of our conclusions, nor any of the statistical tests described in the manuscript, are appreciably changed.

A slight update to the analyses depicted in **Figure 3**, and the accompanying **Supplementary Figs. 7-8**, should be noted. During the course of the update to antiSMASH 5, we noticed that version 5 of this platform occasionally outputs very small predicted structures in which a small and ubiquitous backbone is attached to an unspecified R-group, using the SMILES token `[*]`. Two examples, shown below, correspond roughly to "unspecified amino acid" and "unspecified malonyl unit". In our view, these predictions are sufficiently small and vague that they do not embody any actionable information about the chemical structure of the encoded product. Inspecting PRISM predictions, we found that similarly uninformative structures were occasionally output (e.g., a single malonyl unit). Thus, in the interest of fairness, we applied a molecular weight filter to remove structures under 100 Da predicted by either program. This has led to some further minor adjustments to the numbers presented in this section.

We have revised **Figure 2a-d**, **Figure 3**, and **Supplementary Figs. 4a, 7, and 8** to reflect the update from antiSMASH 4 to 5. Slight revisions have also been made to the number of clusters for which structures were predicted by antiSMASH 5 and/or PRISM 4 throughout the Results section.

The source code for PRISM4 is not accessible via the link mentioned in the manuscript. Many academic journals have a policy in place for making the scientific software available to users without making them register on such servers. The current implementation of PRISM4 mandates acceptance of NFP EULA and registration of academic users. This might restrict many researchers in

using this tool, and overall community acceptance of these novel methods. The editorial policy of N.COMM in this aspect is not clear.

As noted in our response to a similar comment by reviewer #2, the source code of PRISM 4 itself is central to Adapsyn's commercial viability. For this reason, we regret that it simply is not possible to provide this code to the general public. However, we believe that the provision of PRISM 4 to the community as a freely available resource provides a tremendously useful resource to the community. We also want to clarify that this service carries no cost to users: Adapsyn bears the expense of all necessary computing resources.

However, in keeping with the *Nature Communications* editorial policy, we enclose the PRISM 4 source code itself with the revised manuscript for confidential peer review, to allow its assessment only by the reviewers. This code is included with the resubmitted manuscript, with the understanding it will not accompany publication. We have also provided the Python code used to conduct the analyses presented in the paper via GitHub. In combination with the data already provided in the supplementary information, and newly provided via Zenodo in response to comments from reviewer #2, we are pleased to note that this code will allow the interested reader to replicate every figure in the paper, ensuring the reproducibility of our results.

We have revised the Code Availability statement accordingly as follows:

Code availability. ~~The algorithm described in this paper is available from <https://grid.adapsyn.com/prism>.~~ Source code used to conduct the analyses described in the manuscript is available from GitHub (<https://github.com/Adapsyn/prism-4-paper>).

Figure2: Information in plot c and e should ideally tally. The highest median Tanimoto coeff. in plot c is less than 0.6, while for at least 7 BGC families this value is greater than 0.6.

We apologize for the confusion on this point. The distributions shown in **Figure 2e** reflect varying numbers of BGCs in the gold standard set for each biosynthetic family; as an example, there are many more type 1 polyketides known than antimetabolites. Thus, the distribution of median Tanimoto coefficients shown in **Figure 2c** is essentially a *weighted* sum of the distributions for each individual biosynthetic family, with more abundant families dominating the overall distribution. This accounts for the discrepancy noticed by the reviewer. To clarify this point, we have added the number of clusters of each family in the 'gold standard' set directly into **Figure 2e**, as reproduced below (note that additional modifications to the figure have been made in response to comments from reviewer #2):

(see Figure 2) e, Median and maximum Tanimoto coefficients between true and predicted structures generated by PRISM 4 for the gold standard set, by biosynthetic family and compared to the median Tanimoto coefficient between predicted structures and non-matched BGCs (“random pairs”). Top, statistical significance of the comparison between median and random Tanimoto coefficients (***, $p < 0.001$; **, $p < 0.01$; *, $p < 0.05$, two-sided t-test). Bottom, number of BGCs from each family in the gold standard set. Box plots show median (horizontal line), interquartile range (hinges), and the smallest and largest values no more than 1.5 times the interquartile range (whiskers) throughout.

GUI had issues with correctly displaying the predicted structures when tested on both Chrome and Safari web-browser.

Some bugs in correctly displaying the colours for a particular BGC type were observed.

We regret that the reviewer experienced these issues when testing the PRISM 4 web server. Unfortunately, it is difficult to debug these precise issues without being able to reproduce the exact error the reviewer encountered. However, to address this point, we have developed a new graphical user interface (GUI) ‘skin’ for the PRISM web application. All of the visualizations and functionality from the existing GUI have been carried over, but with a slightly altered graphical style.

Our motivations for developing this new ‘skin’ were twofold. First, we wanted to remove the dependency of PRISM 4 on the Apache Tomcat server environment, which has been the single biggest cause of downtime for the PRISM web server over the past 5+ years. A complete overhaul of the GUI was necessary to deploy PRISM 4 on a non-Tomcat server, which we anticipate will markedly improve the uptime of the web application. An ancillary benefit is that this shift allowed us to deploy PRISM on a industry-standard cloud computing network, instead of our own server grid, which we anticipate will further improve uptime (e.g., as a result of power outages, etc.). Second, as a direct consequence of this update, all of the graphical elements in PRISM (e.g., the colours for BGC types and predicted structures, noted by the reviewer) are now rendered directly in the browser using widely used, third-party libraries such as Vue.js and Material Design. These highly robust libraries form the basis for an enormous number of interactive web applications on the Internet today, and consequently they have been thoroughly usage-tested under far more platforms (e.g., operating systems, browsers) than we could ever realistically test alone. As a result, we expect that the updated GUI will be much more robust to bugs like those the reviewer reports.

Thus, although we were not able to directly evaluate the bugs reported by the reviewer, we believe the new PRISM 4 'skin' will address these issues, in addition to improving the robustness and responsiveness of the web application itself more generally and increasing server uptime. We reiterate that all of the visualizations and functionality from the previous release are present in this new version (and the back-end of PRISM itself has been untouched), but with a new appearance that reflects a shift towards industry-standard libraries for interactive web application design. We have revised the Methods and Supplementary Note to update references to Apache Tomcat:

Methods:

Overview of PRISM 4

PRISM 4 is a **cloud-based**, interactive web application, **with a back-end** written in the Java programming language. The web application **itself consists of a VueJS front-end, paired with a Python API that distributes submissions to background workers, and** is available at <http://prism.adapsyn.com>. A number of steps have been taken to ensure the high performance of the web application, including **horizontal distribution of individual PRISM runs over the cloud**, as well as optimization of key bottlenecks to reduce the runtime by approximately an order of magnitude over PRISM 3 (ref. 29).

Supplementary Note:

PRISM 4 web application

Overview. PRISM 4 is a Java 7 web application, ~~designed for deployment on an Apache Tomcat 7 web server, and is~~ freely available as an online service for the research community at <http://prism.adapsyn.com>. The PRISM web application is powered by Vue.js with a lightweight Python Flask API using PostgreSQL and Redis for queue management, providing a scalable solution that can process many submissions at once. ~~The web application back-end is distributed over a 288-core server grid, ensuring high capacity, with details about a submission's position in the grid queue displayed to users.~~

Limitations of the structure prediction method may be discussed in more details.

We agree with the reviewer's suggestion that readers would benefit from a more detailed discussion of the limitations of our method. To address this comment, we have added a paragraph to the Discussion section describing the chief limitations of PRISM 4, as well as some perspectives on future directions for the field. This text is reproduced below:

Some limitations should be noted. In developing PRISM 4, we set out to codify an enormous corpus of knowledge, accumulated over decades of research in biosynthesis and enzymology, into an algorithmically tractable form. An inevitable consequence of this approach is that PRISM relies on homology between newly detected proteins and known enzymatic machinery in order to reveal BGCs and predict the structures of their genetically encoded products. For this reason, PRISM can neither identify BGCs from undescribed families, nor predict novel enzymatic activities. More generally, current models of secondary metabolite biosynthesis are incomplete, which places an inherent limit on the accuracy of structure prediction; we have sought to address this by revising the systems used for BGC detection and structure prediction as new information has become available. Recently, we and others have shown that deep learning-based methods can enable more flexible and accurate detection or characterization of BGCs or individual biosynthetic components^{27,28}. However, at present these approaches still rely on interfacing with rule-based systems such as that employed by PRISM 4 to permit structure prediction²⁷, or else are not capable of generating predicted structures²⁸. In the future, more sophisticated machine-learning approaches might enable the end-to-end prediction of encoded small molecules directly from primary sequence. Finally, PRISM 4 was designed primarily for prokaryotic genome analysis and

thus cannot identify BGCs families thought to be specific to eukaryotes, and—like all tools for genome annotation—may produce incongruous results when applied to fragmented or low-quality genome assemblies.

Figure 3b and 4b depicts Firmicutes, Firmicutes_A, Firmicutes_B and Firmicutes_C phylum. These phyla names are according to which taxonomy, it is not clear.

We apologize that this was unclear, and have revised both figure legends to clarify that these represent taxonomic classifications from the Genome Taxonomy Database (GTDB), e.g. in **Figure 3**:

(see Figure 3) a–b, Number of BGCs with at least one chemical structure predicted by PRISM 4, antiSMASH, or both methods in a collection of 3,759 dereplicated complete bacterial genomes, by biosynthetic family **(a)** and phylum of producing organisms **(b)**, as classified in the Genome Taxonomy Database (GTDB)¹⁵.

REVIEWERS' COMMENTS

Reviewer #2 (Remarks to the Author):

The authors have satisfactorily addressed all my comments. I recommend publication of the manuscript.

Reviewer #3 (Remarks to the Author):

Thanks for the detailed response. I think the manuscript improved. However, I still have some issues with it.

1) Though the PRISM4 source code was shared for private review, due to absence of instructions to run and test the programs/scripts, it was not possible to provide any comments in this regard. Additionally, in absence of release of the source code, it is not clear to this reviewer, in the event of a bugfix or code improvement done in the backend by the developers after the publication of this manuscript, how the user can ensure a consistent version of the results they receive. This can lead to confusions and the user has no option to trace or confirm this or resolve this ambiguity. Probably this calls for seeking a declaration from the developers about version integrity and transparent release of information. This reviewer is not sure if a third party mechanism is available for code integrity confirmation or if this issue can be handled more professionally.

Regarding the previous comment, "GUI had issues with correctly displaying the predicted structures when tested on both Chrome and Safari web-browser.", we rechecked if the issue was resolved. Unfortunately, there seems to be some technical issue which keeps giving error on the tested Chrome and Firefox web browsers. Error log (Failed to load image: 1 / 50. Please view here to allow for small molecule rendering) [Attached screenshot1]. Error continued even after following the suggested instruction for small molecule rendering.

[Attached screenshot2] Some bugs in correctly displaying the colours for a particular BGC type were again observed.

Reviewer #2 (Remarks to the Author):

The authors have satisfactorily addressed all my comments. I recommend publication of the manuscript.

We sincerely thank the reviewer for the time they took to improve our manuscript.

Reviewer #3 (Remarks to the Author):

Thanks for the detailed response. I think the manuscript improved. However, I still have some issues with it.

1) Though the PRISM4 source code was shared for private review, due to absence of instructions to run and test the programs/scripts, it was not possible to provide any comments in this regard.

We thank the reviewer again for their feedback on the manuscript and their additional comments. Unfortunately, PRISM has always been designed to be deployed as a web application—historically on an Apache Tomcat server, and now, with the release of PRISM 4, in conjunction with a PostgreSQL and Redis back-end. A consequence of this design is that deploying PRISM to run on one's own web server requires some technical expertise. In other words, it is not really possible to provide a version of the source code that the reviewer would be able to test locally without being able to provide detailed instructions and troubleshoot the configuration of their web server. Indeed, this is another reason we feel that hosting the web application ourselves, via the publicly available web server, provides a valuable service to the community.

Additionally, in absence of release of the source code, it is not clear to this reviewer, in the event of a bugfix or code improvement done in the backend by the developers after the publication of this manuscript, how the user can ensure a consistent version of the results they receive. This can lead to confusions and the user has no option to trace or confirm this or resolve this ambiguity. Probably this calls for seeking a declaration from the developers about version integrity and transparent release of information. This reviewer is not sure if a third party mechanism is available for code integrity confirmation or if this issue can be handled more professionally.

We thank the reviewer for highlighting the important issue of version control. First, we would like to offer some reassurances that we have indeed considered this issue. JSON files output by PRISM contain the exact version number, under the 'version' key (for example, all of the files in our Zenodo upload contain the key-value pair "version": "4.3.5"). This allows users to unambiguously associate a particular results file with the version of the underlying PRISM web application, addressing one of the reviewer's concerns. Moreover, the web application will display a warning if a user attempts to re-open a saved JSON file after an update to the PRISM web application (see the attached screenshot below). As a final step to address the reviewer's concerns, we have also added a changelog to the web server, under the 'About' page. This changelog will be used to record any future changes to PRISM, including bug fixes, new features, etc.

There may be incompatibility issues when displaying this JSON report. Current version 4.4.4, JSON report version 4.3.5

Response Figure 2.1. Error message displayed if a user attempts to open a JSON file generated with a different version of PRISM.

Regarding the previous comment, "GUI had issues with correctly displaying the predicted structures when tested on both Chrome and Safari web-browser.", we rechecked if the issue was resolved. Unfortunately, there seems to be some technical issue which keeps giving error on the tested Chrome and Firefox web browsers. Error log (Failed to load image: 1 / 50. Please view here to allow for small molecule rendering) [Attached screenshot1]. Error continued even after following the suggested instruction for small molecule rendering. [Attached screenshot2] Some bugs in correctly displaying the colours for a particular BGC type were again observed.

The screenshots provided by the reviewer indicate that these bugs were encountered on the old PRISM graphical user interface (GUI), and not the new GUI which we developed for the revised manuscript. We had left the old GUI up to allow the reviewers a chance to compare the old and new interfaces, and we deeply regret this confusion this caused. We have taken down the old PRISM GUI, and configured the previous URL (grid.adapsyn.com/prism) to redirect to the new one (prism.adapsyn.com). We would like to take the opportunity to re-emphasize that despite the inconvenience caused, these changes have allowed us to distribute PRISM over a Google Cloud computing backend instead of our own in-house server grid (substantially increasing the throughput of the web application) and move away from its longstanding dependency on Apache Tomcat (substantially increasing uptime). Additionally, we obtained the sequences the reviewer used from the accession numbers shown in the screenshot, and confirmed that we cannot reproduce these bugs in the new GUI.